# On the limitations in assessing stability of oxygen evolution catalysts using aqueous model electrochemical cells

Julius Knöppel [1,2 ✉], Maximilian Möckl[3], Daniel Escalera-López[1], Kevin Stojanovski[1,2], Markus Bierling[1,2], Thomas Böhm[1,2], Simon Thiele [1,2], Matthias Rzepka[3] & Serhiy Cherevko [1 ✉]

Recent research indicates a severe discrepancy between oxygen evolution reaction catalysts dissolution in aqueous model systems and membrane electrode assemblies. This questions the relevance of the widespread aqueous testing for real world application. In this study, we aim to determine the processes responsible for the dissolution discrepancy. Experimental parameters known to diverge in both systems are individually tested for their influence on dissolution of an Ir-based catalyst. Ir dissolution is studied in an aqueous model system, a scanning flow cell coupled to an inductively coupled plasma mass spectrometer. Real dissolution rates of the Ir OER catalyst in membrane electrode assemblies are measured with a specifically developed, dedicated setup. Overestimated acidity in the anode catalyst layer and stabilization over time in real devices are proposed as main contributors to the dissolution discrepancy. The results shown here lead to clear guidelines for anode electrocatalyst testing parameters to resemble realistic electrolyzer operating conditions.

[1] Forschungszentrum Jülich GmbH, Helmholtz Institute Erlangen-Nürnberg for Renewable Energy (IEK-11), Erlangen, Germany. [2] Department of Chemical and Biological Engineering, Friedrich-Alexander-Universität Erlangen-Nürnberg, Erlangen, Germany. [3] ZAE Bayern, Electrochemical Energy Storage, Garching, Germany. ✉email: j.knoeppel@fz-juelich.de; s.cherevko@fz-juelich.de

Global warming is driving the transition from fossil fuels to renewable energies. To support the transfer, economically promising alternatives to petrochemical processes for all sectors have to be established[1]. Owing to its high energy density, low chemical complexity, and high efficiency, hydrogen is among the best candidates for energy storage and distribution[2–4]. If hydrogen production from water electrolysis (WE) is fully supplied by renewable energy sources, greenhouse gas emissions can be reduced by 75%[5]. Therefore, research funding on upscaling WE technology is increasing[6,7]. Currently, technologies, based on liquid alkaline and acidic solid electrolyte are equally considered. Classical alkaline electrolyzers lack the option of dynamic operation necessary for direct coupling to fluctuating energy sources, and solid electrolyte anion exchange membrane (AEM) electrolyzers are not at a technology readiness level suitable for upscaling[8,9]. Acidic proton exchange membrane (PEM) electrolyzers consisting of membrane electrode assemblies (MEA), which lack these disadvantages, are the preferred system for upscaling in the short term[10].

It is generally accepted that acidic conditions and high potentials at the anode side of PEM water electrolyzers (PEMWEs) where the oxygen evolution reaction (OER) takes place, demand for materials with high catalytic activity, and corrosion stability. Such criteria are only satisfied for electrocatalysts based on scarce noble metals such as iridium (Ir) and, to a lesser extent, ruthenium (Ru). Although their current implementation in PEMWEs is not hampered by their cost, upscaling fabrication of MEAs with these metals is expected to be a major cost driver at the GW scale[11,12]. Therefore, a significant part of research activities on WE are focused on the reduction of noble metal content in PEMWE anodes[13,14].

For the state-of-the-art Ir catalysts, a cornerstone in fundamental research studies has been to maximize Ir utilization, specifically, to increase their OER mass activity whilst reducing noble metal content without a significant loss in activity[15,16]. The use of high-surface-area catalyst supports[17,18], highly active perovskites[19,20], and multimetallic materials[21,22] are employed to reduce the noble metal content. However, stability has to be monitored as a second major descriptor in electrocatalyst design and synthesis, as OER catalysis also triggers catalyst dissolution[23].

Activity and stability evaluations of newly developed catalysts are performed ex situ in the classical three-electrode electrochemical cell setup with acidic electrolyte to simulate the acidic pH environment of PEMWEs anodes in presence of Nafion[24–26]. Current-potential profiles are recorded and analyzed for activity evaluation. For stability evaluations, however, more sophisticated methods were developed, such as electrochemical quartz crystal microbalance[27], scanning flow cell (SFC) coupled to an inductively coupled plasma mass spectrometer (SFC-ICP-MS)[28,29], and post-analysis of electrolyte and catalyst layers[30,31]. However, comparative data of catalyst stability in both systems show that degradation in aqueous systems does not represent the conditions in PEMWE[32,33].

Recent results from our group, based on aqueous SFC-ICP-MS measurements and end-of life data from PEMWE, indicate an underestimation of the actual catalyst lifetime of several orders of magnitude, ranging from days in aqueous to years in MEA[28]. In this work, the relevance of S-numbers, a new metric for OER catalyst lifetime estimation, measured in aqueous systems for real application based on end-of-life data of PEM electrolyzers, is discussed. As end-of-life data is rarely found for Ir-based electrolyzers, the data set of a system using $RuO_2$ as anode material published by Ayers et al.[14] was compared with the same material measured in SFC-ICP-MS. It was found that the PEM electrolyzer outperforms the aqueous system by about three orders of magnitude, leading to a significant increase in the estimated lifetime of the electrolyzer in comparison to the same catalyst in the aqueous system. A similar concept, the activity-stability factor, was developed in parallel by Kim et al.[34].

In this work, we aim to reveal the experimental factors responsible for the observed OER catalyst dissolution differences between aqueous model systems (AMS) and PEMWEs. We evaluated how the parameters that diverge between the systems such as catalyst loading, mass transport conditions, Nafion binder content, and electrolyte pH influence Ir dissolution. Also, we aim to determine the real dissolution rates of MEAs for PEMWEs using a custom-made full cell setup devised to prevent galvanic precipitation of catalyst dissolved species under OER operation.

## Results and discussion

**Iridium OER catalyst dissolution: aqueous model versus MEA systems**. The dissolution behavior of OER catalysts in AMS is already well studied[28,35–37]. Utilizing online measurements, the dissolution behavior of OER catalysts under various electrochemical conditions has been shown. To put the results presented in this section into context, it is important to highlight the commonalities and differences between MEA and AMS. In AMS, the employed electrolyte, mostly an acid or base, is diluted by the reactant, deionized (DI) water. As schematically shown in Fig. 1a, the reaction products, $H_2$ at the cathode side and $O_2$ and protons at the anode side, as well as dissolution products, such as $Ir^{3+}$,

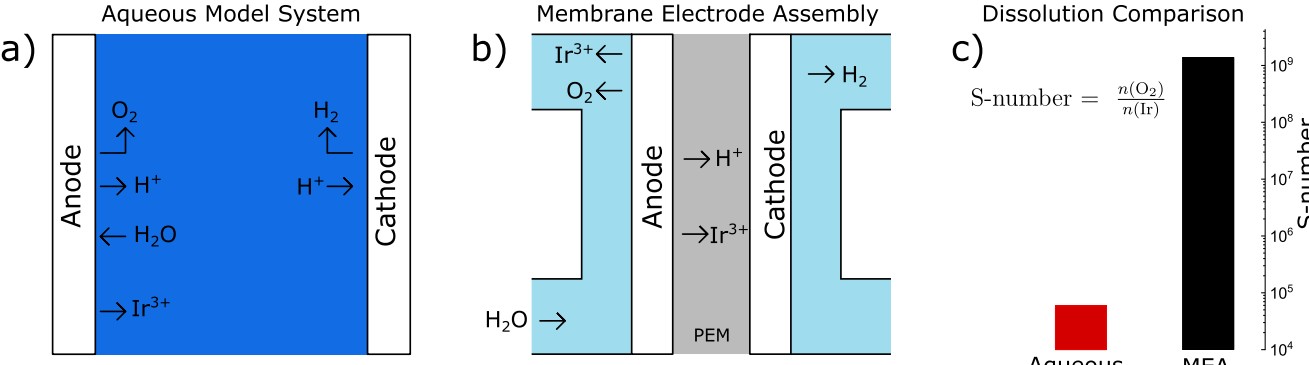

**Fig. 1 Degradation processes of OER catalysts in aqueous and polymer electrolyte. a** Schematic drawing of degradation processes in a classical aqueous electrolysis cell in aqueous electrolyte and **b** schematic drawing of degradation processes in MEA. **c** Dissolution stability of $IrO_x$ under OER conditions in aqueous electrolyte, measured in SFC-ICP-MS, and polymer electrolyte, measured in a precipitation-free MEA device, expressed in the S-number metric. Measurements were carried out with a 5 min chronopotentiometry hold in aqueous electrolyte and over several days for MEA, typical timescales for the devices. Source data are provided in the source data file.

diffuse into the bulk. Hence, to measure dissolution, it is sufficient to take aliquots of electrolyte from the solution. Utilizing flow cells, which directly transport reaction products from the reaction site downstream (if coupled) to analytical techniques such as ICP-MS, the dissolution behavior can be directly correlated to the electrochemical operation[28,34,38].

To study the degradation behavior of MEA, a system with a much higher degree of complexity, long-term measurements, and end-of-life (EOL) data have been used thus far. Owing to the long lifetime of MEA electrolyzers, however, EOL data are scarce. Furthermore, measurements of dissolution products in MEA are more complicated than in AMS. As schematically shown in Fig. 1b, electrolyte and reactant are decoupled in MEA by placing the polymer electrolyte between the electrodes and circulating DI water as the reactant at the backside. Reaction products, $H_2$, and $O_2$ escape through porous transport layers at the respective electrodes, whereas $H^+$ is transported through the PEM towards the reaction site at the cathode. In this system, dissolution products of OER catalysts have two ways to escape the anode catalyst layer: through the anode water cycle or the membrane towards the cathode side. Furthermore, galvanic replacement (GR) of dissolution products with stainless steel tubes, often employed in MEA test setups, can lead to an underestimation of dissolution[39].

Hence, to reliably determine dissolution in MEA and realistically compare results with AMS, several factors have to be controlled. The water level in the anode compartment and the water flow at the cathode outlet through electroosmotic drag have to be monitored at all times[40,41]. Furthermore, the amount of iridium depositing in the membrane has to be estimated. Also, GR should be excluded as a measurement factor.

For such purpose, a dedicated MEA setup without metallic parts in the anode water cycle was developed and employed in this study. Cell components involved in electronic conduction, namely titanium flowfields and current collectors, are coated with gold or platinum to prevent GR. Samples are taken from the anode water cycle and the cathode outlet and analyzed separately by ICP-MS. The setup and flow scheme is shown in supplementary note 1. Water balance calculations, necessary to determine the amount of dissolved iridium, are shown in supplementary note 2.

To compare the dissolution stability of OER catalysts between MEA and AMS, a commercially available $IrO_x$ catalyst is measured in the aforementioned dedicated MEA system as well as in an SFC-ICP-MS setup operated with 0.1 M $H_2SO_4$. Figure 1c shows dissolution stability in both systems, displayed in the S-number metric, a dimensionless descriptor that compares the amount of oxygen evolved, calculated from the measured current density at an estimated 100% faradaic efficiency towards OER, with the amount of iridium dissolved $\left( \text{S} - \text{number} = \frac{n(O_2)}{n(Ir)} \right)$[28]. S-numbers were calculated from constant current measurements of 5 minutes in AMS and several days in the MEA. Electrochemical data and dissolution data for MEA experiments are available in supplementary note 3. Electrochemical data, dissolution data, and the integration areas for determining S-numbers in SFC-ICP-MS are shown in Fig. 2. Both timescales are representative of the respective system.

The S-number of $IrO_x$ in the SFC-ICP-MS is $6 \times 10^4$. Although with the used $H_2SO_4$ electrolyte, a stronger adsorption of anions on the surface is anticipated[42], the measured S-number is comparable to literature values measured in the non-coordinating $HClO_4$, which range between $10^4$ and $10^5$[28,43–45]. Thus, although the influence of the electrolyte anion cannot be fully ruled out, its role in the stability of $IrO_x$ is minor. Remarkably, the observed S-number of $IrO_x$ in the MEA system

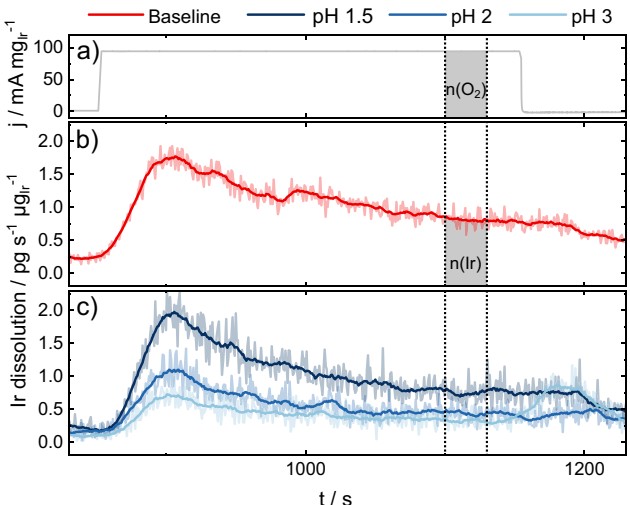

**Fig. 2 Dissolution profiles of $IrO_x$ catalyst spots in SFC-ICP-MS measurements. a** Applied current step and **b, c** resulting dissolution of **b** a baseline measurement at standard conditions (10 $\mu g_{Ir}$ cm$^{-2}$ catalyst loading, 200 $\mu l$ min$^{-1}$ flow rate, fresh electrolyte, 33 wt% Nafion in the catalyst layer and 0.1 M $H_2SO_4$ (pH = 1)) and **c** with a variation of electrolyte pH. The integration area for the calculation of S-numbers is highlighted by vertical dashed lines. Source data are provided in the source data file.

exceeds the one observed in aqueous systems by almost five orders of magnitude. Further experiments in AMS were undertaken to unravel the reasons for this behavior.

**Evaluation of model aqueous OER stability parameters**. We evaluated several parameters that generally differ in both systems to determine the origin of the dissolution discrepancy between AMS and MEA. (a) catalyst loading; (b) electrolyte flow rate; (c) presence of electrochemically dissolved iridium species; (d) Nafion content in the catalyst layer; and (e) pH were individually varied during testing in the AMS (SFC-ICP-MS). All experiments were carried out with the same $IrO_x$ catalyst powder as in previously shown MEA experiments.

Dissolution profiles of baseline measurement of $IrO_x$ catalyst powder spots at a current step of 100 mA $mg_{Ir}^{-1}$ (Fig. 2a) are displayed in Fig. 2b. The full measurement protocol is shown in supplementary note 4. Representative electrochemical and dissolution data for all experiments are shown in supplementary note 5.

The corresponding S-numbers for all experiments, determined at a 30 s steady-state interval at the end of the current step[28], as shown in Fig. 2a and b) are displayed in Fig. 3. All error bars are acquired from at least three independent measurements. The baseline measurement is hereby displayed in Fig. 3a).

Our first study focused on differences in catalyst loading. Although Ir loading in aqueous studies rarely exceeds 10 $\mu g_{Ir}$ cm$^{-2}$, loading in MEA is typically ~1–2 $mg_{Ir}$ cm$^{-2}$[46]. S-numbers of catalyst spots with different loading is shown in Fig. 3b, where loading is varied between 10 $\mu g_{Ir}$ cm$^{-2}$ and 250 $\mu g_{Ir}$ cm$^{-2}$. The obtained S-number values are comparable and in the same order of magnitude. Hence, we can exclude the influence of loading on the dissolution discrepancy.

The second study was different SFC operating flow rates. Given the flow rate uncertainty in PEMWE MEA systems, contrasting with its precise control in our SFC-ICP-MS setup, we evaluated the impact of SFC flow rate by variations within one order of magnitude. The flow rate of electrolyte to the ICP-MS was here varied between 66 $\mu l$ min$^{-1}$ and 740 $\mu l$ min$^{-1}$. S-numbers for these experiments are virtually equivalent as shown in Fig. 3c. As

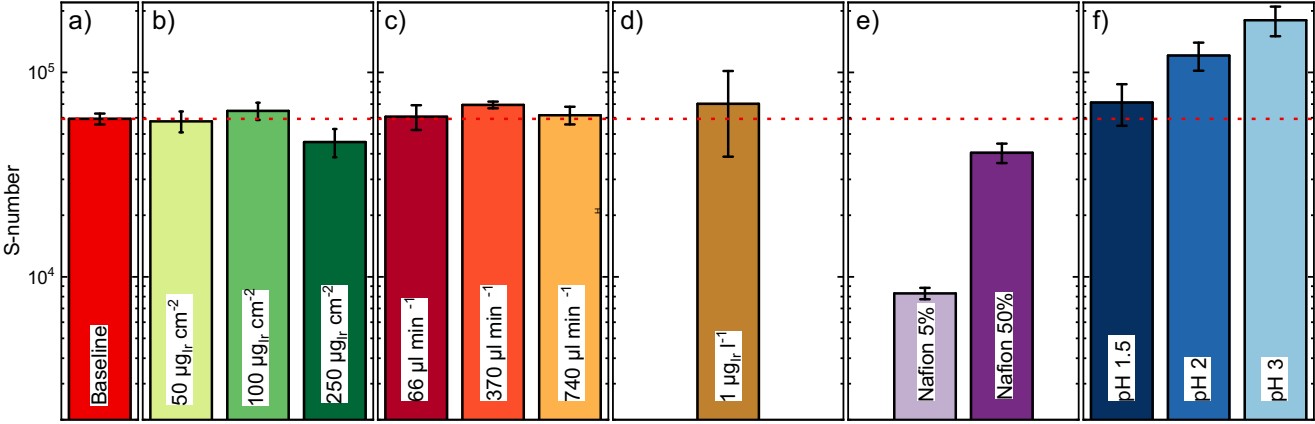

**Fig. 3 S-numbers of IrO$_x$ catalyst spots with varied conditions, measured in SFC-ICP-MS. a** Baseline measurement. **b** Variation of loading. **c** Variation of flow rate. **d** Addition of pre-dissolved iridium in the electrolyte. **e** Variation of Nafion content in the catalyst layer. **f** Variation of pH. Red, dashed line: baseline measurement (10 µg$_{Ir}$ cm$^{-2}$ catalyst loading, 200 µl min$^{-1}$ flow rate, fresh electrolyte, 33 wt% Nafion in the catalyst layer and 0.1 M H$_2$SO$_4$ (pH = 1)), displayed for ease of comparison between **b** and **f**. Error bars were calculated as the standard deviation of at least three independent measurements. Source data are provided in the source data file.

the flow rate effectively dictates the mass transport rate of dissolved species from the electrode interface to the bulk electrolyte, a possible Ir redeposition mechanism should be affected by it as well. Based on the results obtained, these mass transport phenomena do not seem to play a key role in the AMS-MEA catalyst stability differences.

The third parameter variation tackled the dissolution–redeposition equilibrium of Ir-dissolved species, which can occur during MEA water circulation owing to low flow rates and slow mass transport in thick catalyst layers. To simulate such conditions in AMS, and enable a potential dissolution–redeposition mechanism, electrochemically dissolved iridium has to be present in the electrolyte in relevant concentrations. Therefore, Ir was electrochemically dissolved from polycrystalline iridium[37,47] (see the experimental section for further details) and then intentionally incorporated into the acidic electrolyte used. The iridium concentration in the electrolyte is determined by ICP-MS and adjusted accordingly. Figure 3d) shows the S-number of IrO$_x$ spots measured with 1 µg$_{Ir}$ l$^{-1}$ dissolved iridium in the electrolyte. This value is similar to a recent publication, where Ir concentrations in MEAs were measured[39]. Also, it is in the same order of magnitude as the concentrations measured in our MEA study (see supplementary note 3). As observed with the previous parameters, S-number values are comparable to the baseline experiment. Hence, equilibrium states between dissolution and redeposition are unlikely to largely contribute to the Ir dissolution discrepancy.

The fourth evaluated parameter is the influence of Nafion content in the catalyst layer on Ir dissolution. Unlike OER rotating disk electrode experiments, where no Nafion is required, it acts as a catalyst layer binder in SFC-ICP-MS to avoid particle detachment. Figure 3e shows the S-numbers of IrO$_x$ spots with different Nafion contents, varied here between 5 and 50 wt% vs. the total catalyst content. Interestingly, the dissolution rate of catalysts spots with 5 wt% Nafion in the catalyst layer is significantly larger from the other Nafion contents. Indeed, the stability differs from baseline measurements by an order of magnitude, with an S-number of 8 × 10$^3$. We hypothesize that, at lower Nafion contents, dissolved iridium mass transport from the catalyst layer to the electrolyte bulk might be more efficient. A local saturation might hereby inhibit dissolution. As baseline measurements have a Nafion content of 33 wt%, whereas MEA electrodes were produced with 9 wt% Nafion content in this study, Nafion content will have a role in stability observations.

However, different constraints have to be met for measurements in both systems. In SFC-ICP-MS, Nafion predominantly acts as a binder in the catalyst layer to avoid particle detachment. On the other hand, in the more complex MEA system, Nafion has not only an influence on the integrity of the catalyst layer, but also on features such as contact resistance with the porous transport layers (PTL). As Nafion contents employed in this study rather open the gap between AMS and MEA instead of closing it, this rather leads to an underestimation than an overestimation of the stability discrepancy.

The fifth study carried out is the variation of the working electrolyte pH value, varied here between 1 and 3 (experimental dissolution profiles shown in Fig. 2c). To ensure a similar ionic strength, a sulfate salt was added to electrolytes with pH>1. The resulting S-numbers, shown in Fig. 3f), significantly differ from each other. Although the S-number metric obtained for pH 1.5 is still similar to the reference measurement (pH 1), a significant difference is observed for pH 2 and 3. Such difference in stability is a factor of three: for pH 1, S-number = 6 × 10$^4$ whilst for pH = 3, S-number = 1.8 × 10$^5$. It should be noted that local pH at the electrode under OER conditions might be lower owing to the worsened buffer capacity of electrolytes with higher pH[48].

MEA environments for PEMWE are, according to the literature, highly acidic owing to the use of Nafion[49,50]. The observations, made here in AMS, however, indicate that the activity of protons in MEA electrolysis might be lower than generally accepted. Further experiments in MEA systems have to be conducted to unravel the magnitude of the differences.

**Impact of pH in PEMWE operation: catalyst and MEA stability.** As shown in a previous section, of all parameters evaluated in our study of model aqueous systems, pH is the only one with a relevant impact on stability towards closing the gap. Hence, measurements of IrO$_x$ in MEA were conducted with 0.1 M H$_2$SO$_4$ in the anode water cycle to investigate the influence of low pH environments on dissolution. The operational principles in both systems are schematically shown in Fig. 4a for MEA operated with DI water and in Fig. 4b) for the MEA operated with acid. S-numbers from liquid samples collected after 2 h operation are displayed in Fig. 4c). Strikingly, the S-number of the conventional system exceeds the one from the acid operated by more than two orders of magnitude. In comparison to AMS, which operates with

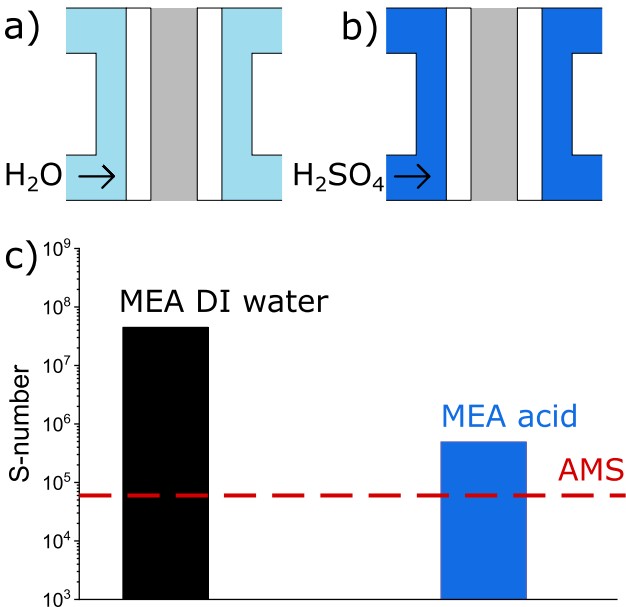

**Fig. 4 Comparison of stability in different systems. a** Working principle of an MEA operated with DI water. **b** Working principle of an MEA operated with 0.1 M $H_2SO_4$. **c** S-numbers of MEAs operated with DI water and 0.1 M $H_2SO_4$ after 2 h of measurement. The red, dashed line in **c** indicates the S-number of the AMS system. Source data are provided in the source data file.

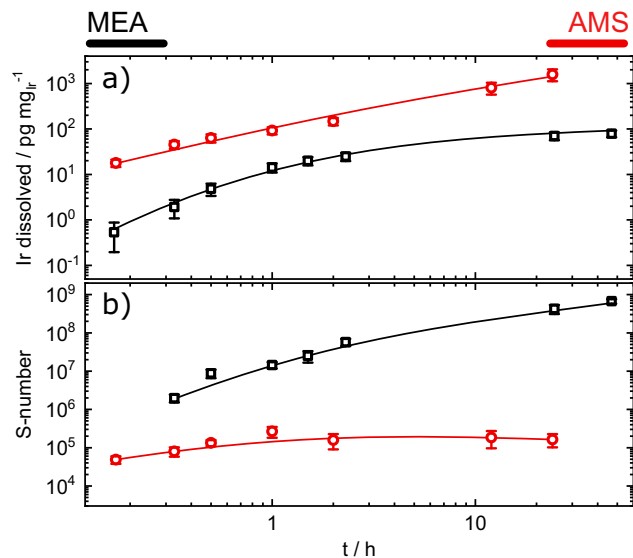

**Fig. 5 Long-term stability of $IrO_x$ in AMS and MEA environment.**
**a** Loading-normalized total dissolved iridium amount at current densities of 0.2 A $mg_{Ir}^{-1}$ and 2 A $mg_{Ir}^{-1}$ in AMS and MEA, respectively; **b** S-numbers calculated from the amounts of dissolved iridium. Lines displayed for ease of interpretation. Error bars were calculated as the standard deviation of at least three independent measurements. Source data are provided in the source data file.

S-numbers around $6 \times 10^4$, the stability difference to the acidic operated MEA virtually vanishes. Its impact can be easily grasped when calculating catalyst half-life estimated from S-numbers[28]. For a DI water-fed MEA system, its value is ca. 150 years, whereas for an acidified MEA it is just several days.

Post mortem scanning transmission electron microscopy (STEM) cross-section micrographs of the MEA after 48 h of continuous operation at 2 A $mg_{Ir}^{-1}$, shown in supplementary note 7, reinforce the stark degradation differences found from liquid sample analysis. Although the anode catalyst layer of the DI water-operated MEA is virtually intact and iridium migration into the membrane is non-existent, the anode catalyst layer of the acidic operated MEA reveals exceptional signs of degradation. Indeed, Au particles, originating from the partly dissolved flowfield coating, of µm diameter form in the membrane close to the anode side. The cathode side of the conventional operated MEA only shows signals of Pt and C, whereas iridium was detected in the catalyst layer of the acidified MEA. Furthermore, the MEA polarization curves indicate a shift in pH (for full description, see supplementary note 8)[51] and the anode flowfield and current collector displays stark signs of degradation after the experiment. (See supplementary note 9)

**Impact of OER operating timescale in catalyst stability in AMS and MEA.** The different timescales have to be taken into account when comparing AMS with MEA systems. For instance, the data shown in Fig. 1c were measured over minutes for AMS and days for MEA. Indeed, MEA systems have proven stable operation for thousands of hours on the laboratory scale[52] and in industrial applications[53]. Hence, we investigated if the short experimental timescale in AMSs can be extrapolated to PEMWE systems, or stabilization effects occur over large timescale operating conditions.

SFC-ICP-MS measurements cannot be carried out for several hours or even days. Thus, electrochemical measurements were carried out in an H-cell configuration to represent an aqueous model system operated at longer timescales. A 0.1 M $H_2SO_4$ electrolyte with sample collection with sample collection from both compartments was used (see experimental). To have a side-by-side comparison, samples from the developed GR-free MEA system were taken from the anode water cycle and the cathode water outlet. Calculations for obtaining mass losses are shown in supplementary note 2.

Loading-normalized iridium dissolution for H-cell ($IrO_x$ loading = 10 $\mu g_{Ir}$ $cm^{-2}$) and MEA measurements (anode $IrO_x$ loading = 1 $mg_{Ir}$ $cm^{-2}$) are displayed in Fig. 5a. Current densities were 0.2 A $mg_{Ir}^{-1}$ for AMS and 2 A $mg_{Ir}^{-1}$ for MEA. At the employed current density, the amount of dissolved iridium in the aqueous system rose almost constantly throughout the experiment after an early-stage stabilization. S-numbers (Fig. 5b) only stabilized marginally from values of $7 \times 10^4$ to $2 \times 10^5$. In contrast, the iridium mass loss during DI water-operated MEA experiments stabilized rapidly after the start to an almost constant level. The S-numbers rose from an initial value of $10^7$ in the first hours to a value of $10^8$ and stabilize after the first day of the experiment at ~$10^9$ (compare supplementary information 2).

The results obtained in MEA are in heavy contrast with results previously shown by Babic et al.[39]. In their experiment, the authors observed fluctuating or, after an initial increase, decreasing iridium concentrations in the electrolyte. Although a direct protocol comparison is not feasible, a similar decrease of iridium in the water feed was observed by Regmi et al.[54]. Our results clearly show the advantage of the employed metal-free MEA setup, as the authors already pointed out a possible interference of GR in their results. Thus, in all studies concerning dissolution products of operational MEAs, GR should be taken into account, as otherwise, it might result in misleading conclusions.

Previous research would suggest a stabilization under long-term operation on iridium-based catalysts owing to crystallization[30]. Indeed, both aqueous and MEA systems should eventually yield an equivalent degree of stabilization. For MEA

systems, such stabilization is reached at an earlier stage given the higher operating current density of $2 \text{ A mg}_{Ir}^{-1}$ inherent to the system compared with the current density of $0.2 \text{ A mg}_{Ir}^{-1}$ employed in our H-cell setup. Because of the low operational currents achieved in AMS, we cannot unambiguously proof such a stabilization effect. A method to circumvent such limitation, beyond the scope of this report, would be to perform studies on high-current density achieving gas diffusion electrode-type (GDE) reactors[55–57].

**System breakdown of the dissolution discrepancy between AMS and MEA.** A comparison of these results reveals the inherent differences between MEA and AMS. As shown in Fig. 6, the differences in S-numbers between AMS (1) and MEA decrease to less than one order of magnitude when circulating diluted acid through the MEA water feed (2). These differences cannot be related to a sole factor. Loading, flow rate, dissolved species, timescale or Nafion content, although not relevant factors in AMS, might play a role in MEA and have to be addressed in a follow-up study. However, the higher complexity of the MEA does not allow tangible conclusions on the main contributor. When operating an MEA with DI water on the same timescale (3), the S-number increases by two orders of magnitude, indicating a pH shift between AMS and MEA as a main contributor to the dissolution discrepancy. After 2 weeks of operation, the S-number of the MEA increased by more than one order of magnitude (4). At this timescale, based on lifetime calculations from the S-number, the catalyst in the aqueous system would already have been degraded completely. Hence, a stabilization on a larger timescale can be treated as the second main contributor to the dissolution discrepancy. As STEM-energy dispersive X-ray (EDX) only detects small amounts of iridium in the membrane close to the anode catalyst layer, iridium depositing in the membrane can be ruled out as a large contributor.

Given the aforementioned results, we should now address past preconceptions regarding local pH during PEMWE operation. PEMWE anodes are assumed to operate under highly acidic conditions due to protons generated at the anode side during operation. However, literature is scarce regarding actual pH value estimation under PEMWE operation. An initial review by Carmo et al.[10] tentatively estimated a pH 2 value, whereas later investigations reported pH values in the anode and cathode water cycle of an MEA setup in a range between 5.6 and 3.5[58]. However, these results might not be representative for conditions in the anode catalyst layer as the local pH can decrease owing to proton generation in the water-splitting reaction. The results shown in this study indicate that proton activity in MEA environment is supposedly lower as estimated from the concentrations. Thus, it is likely, that the effective pH in MEA environment is less acidic as generally accepted in the literature.

These results indicate that future research should emphasize more on two aspects of MEA development: real conditions in the anode catalyst layer and effects in the catalyst/membrane interface.

**Conclusions and outlook.** In summary, we show that stability measurements performed in AMS have to be treated carefully regarding their relevance for long-term PEMWE applications. The main effects contributing to the dissolution discrepancy were identified as a discrepancy between estimated and real pH in MEA and stabilization occurring over time.

Based on this research, the community should critically evaluate the process of OER catalyst testing in AMS. Owing to faster degradation, AMS might serve as an ideal system for accelerated stress tests. For this purpose, GDE systems currently under development might help to study OER catalysts in model systems, which resemble the conditions in MEA much better.

However, those results should always be critically compared with experimental MEA data to extract representative conclusions. Furthermore, more focus should be put on the direct evaluation of catalysts under MEA device operation. In particular, a setup consisting of an MEA coupled to downstream analytics would allow a better understanding of dynamic operation conditions, relevant for coupling to renewable energies. We believe that the results shown here will provide improved

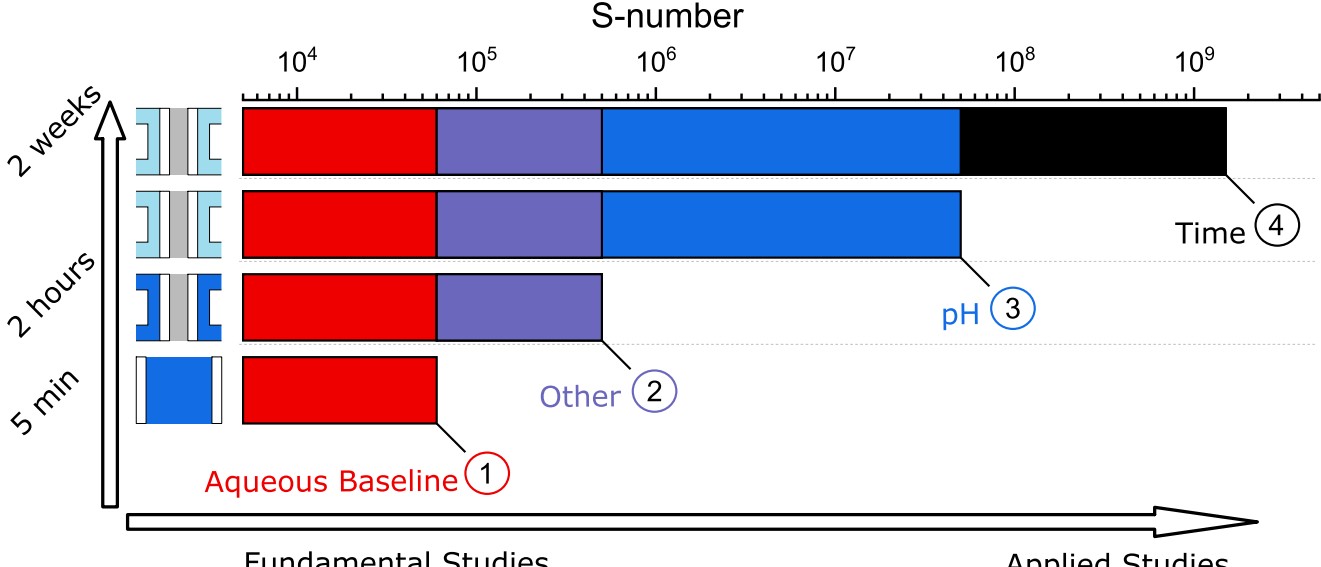

**Fig. 6 Scheme on the proposed main contributors to the dissolution discrepancy.** Schematic drawing of the factors contributing to the OER catalyst dissolution discrepancies between AMS and MEA. Measurements in AMS exhibit an S-number of $6 \times 10^4$ (1). The discrepancy to an MEA operated with acid (2), showing an S-number of $4 \times 10^5$, cannot be pointed to a single factor. Various factors such as flow rate, Nafion content, and timescale have to be taken into account. The discrepancy of the MEA operated with acidic water feed to an MEA operated conventionally with DI water (3) of almost two orders of magnitude, however, is related to a pH shift in the system. With a longer operation time (4), additional stabilization effects in MEA take place. Source data are provided in the source data file.

guidelines for future catalyst development and testing to mimic realistic MEA operating conditions.

## Methods

### SFC-ICP-MS measurements
*Electrode preparation.* Electrodes were prepared by suspending commercial Alfa Aesar $IrO_x \cdot 2H_2O$ Premion catalyst powder in a mixture of 87.5% ultrapure water (Merck Milli-Q), 12.5% IPA, and Nafion® perfluorinated resin solution (Sigma Aldrich, 5 wt%). Standard inks had an iridium concentration of 663 µg l$^{-1}$, a Nafion concentration of 332 µg l$^{-1}$ and a volume of 1 ml in a 1.5 ml Eppendorf tube. Inks were sonicated for 10 min (4 s pulse, 2 s pause) and dropcasted as 0.2 µl on a freshly polished glassy carbon plate (SIGRADUR G, HTW). The quality and diameter of the dropcasted catalyst spots (Ø of ca. 1.3 mm) was screened by employing Keyence VK-X250 profilometer.

For variations of loading and Nafion content, the concentration of iridium and Nafion in the ink was adjusted to the desired loading and concentration. An IPA: DI water ratio of 12.5:87.5, was employed in all $IrO_x$ inks, accounting for the alcohol content in the Nafion solution.

*Electrochemical measurements.* Electrochemical measurements were carried out with an SFC-ICP-MS[29], with the modifications described in ref. [28] in Ar-saturated 0.1 M $H_2SO_4$ (Merck Suprapur) mixed with ultrapure water. The dropcasted spots, acting here as working electrodes, were located with a top view camera to enable vertical alignment with the SFC (Ø 2 mm). A graphite rod served as counter electrode, whereas a saturated Ag/AgCl electrode (Metrohm) was used as reference electrode. ICP-MS measurements were performed with a NexIon 300 spectrometer (Perkin Elmer), employing a flow rate of 208 µl min$^{-1}$ for reference measurements. For the flow rate-dependence studies, flow rates were adjusted by tuning the speed of the ICP-MS peristaltic pump. Daily calibration of the ICP-MS was performed by freshly prepared standard solutions containing Ir (0.5 to 5 µg l$^{-1}$), and Re (10 µg l$^{-1}$) as an internal standard. All current and dissolution rates shown in this report have been normalized to the nominal loading of the spots.

For the variation of dissolved iridium in the electrolyte, iridium was electrochemically dissolved in 0.1 M $H_2SO_4$ by 1000 cyclic voltammograms recorded in a potential range from 0.05 $V_{RHE}$–1.5 $V_{RHE}$[37,47]. The iridium concentration was then determined by ICP-MS. Electrolyte and standards were prepared from the electrolyte with dissolved iridium. A baseline measurement was taken before cell contact.

For the variation of pH, the electrolyte was set to the corresponding $H_2SO_4$ concentration. To ensure electronic conductivity of the electrolyte in measurements with a pH higher than 1, the total concentration of sulfate ions was set to 0.05 M with $K_2SO_4$ (99.999% purity, Sigma Aldrich).

### H-cell measurements
*Electrode preparation.* Electrodes were prepared from Alfa Aesar $IrO_x·2H_2O$ Premion powder. Ink for electrodes was prepared with ultrapure water (Merck Milli-Q) at a concentration of 283 µg l$^{-1}$ with a volume of 1 ml in an Eppendorf tube. The ink was sonicated for 15 min (4 s pulse, 2 s pause) and dropcasted as 10 µl on a freshly cleaned FTO plate, previously sonicated for 10 min sequentially in 2% Hellmanex III (Hellma Analytics) solution, DI water, and ethanol, respectively. The resulting dropcasted Ir catalyst exhibited a diameter of 6 mm and a loading of 10 µg cm$^{-2}$.

*Electrochemical measurements.* Electrochemical bulk measurements were carried out in a homemade H-cell. Each compartment was filled with 28 ml 0.1 M $H_2SO_4$ (Merck Suprapur diluted with Merck Milli-Q) before the experiment. The working electrodes and reference electrodes (Basi, 3 M Ag/AgCl) were immersed in one compartment whereas the counter electrodes (glassy carbon, SIGRADUR G, HTW) were immersed in the other compartment. The compartments were covered with Parafilm to avoid evaporation of electrolytes. Convection in the system for equal distribution of dissolution product was enabled through Ar-purging of the anode compartments. Samples were taken by an automated liquid handler (Gilson GX-271). The electrochemical protocol (Gamry Interface1000 B) was started after the first sample was extracted. The total volume of electrolyte in both compartments was kept between the initial 28 ml and 24 ml at any time.

### MEA measurements
*MEA preparation.* For the experiments with the PEMWE setup, square format 5 cm$^2$ active cell area MEAs were prepared by a decal transfer method. As catalyst for the OER at the anode side, the same Alfa Aesar $IrO_x$ 2$H_2O$ Premion powder was applied as in the SFC and H-Cell experiments. The anode catalyst loading was 1.03 ± 0.07 mg$_{Ir}$ cm$^{-2}$ for all tests. For the hydrogen evolution reaction at the cathode side, carbon-supported (Vulcan XC72) platinum nanoparticles catalyst (45.8 wt% Pt/C; TEC10V50E from Tanaka, Japan) with loadings of 0.30 ± 0.14 mg$_{Pt}$ cm$^{-2}$ was used. To prepare the catalyst inks, catalyst powder, 2-propanol (purity ≥ 99.9 % from Sigma Aldrich, Germany), and Nafion® ionomer solution (20 wt% ionomer; D2021 from IonPower, USA) are mixed for 24 h using a roller mill and 5 mm zirconia grinding balls. The decals were coated with a Mayer-rod coating machine on 50 µm PTFE foil (from Angst+Pfister, Germany). Finally, MEAs were hot-pressed (3 min at 155 °C, 2.5 MPa) using the decals and different

Nafion® membrane types 117 (180 µm thickness), 212 (50 µm thickness) and 211 (25 µm thickness). By evaluating the weight differences (±15 µg; XPE105DR microbalance from Mettler Toledo, Germany) of the PTFE decals before and after hot pressing, the individual catalyst loadings of the MEAs were calculated. All anodes have an ionomer content of 9 wt%, whereas all cathodes have an ionomer to carbon mass ratio of 0.6/1.

*MEA measurements.* To prevent any precipitation of dissolved Iridium in the cell or in the test rig a special PEMWE setup was developed.

*Cell.* The cell uses a two-piece monopolar plate concept consisting of a metal flowfield sheet and a plastic body. The flowfield plates are made from 3 mm grade two titanium sheet with laser cut single serpentine channel (equal 1 mm land and 1 mm channel spacing). To prevent galvanic plating of iridium, the titanium flowfield plates are gold-coated (0.5 µm by physical vapor deposition and another 5 µm galvanic coating on top). Finally, the metal flowfield plates are inserted into a fitted plastic body made from polyoxymethylene (aqueous tests) or polytetrafluoroethylene (second design for aqueous and diluted sulfuric acid tests). The plastic body allows for media transport from the serpentine flowfield inside the cell without contact to metal surfaces to the in- and outlet fittings made from polypropylene (PP) at the face sides of the monopolar plates. At the anode side, an expanded titanium metal sheet (250 µm thickness, Sylatech, Germany) with 5 µm platinum coating is used as a porous transport layer between MEA and flowfield. The cathode side PTL is a carbon fiber paper (TGP-H-120 from Toray, Japan, no MPL) with a thickness of 370 µm.

*Test rig.* A fully automated test rig (E40 by Greenlight Innovation, Canada) equipped with a potentiostat and a booster (Reference 3000 and 30 A booster, Gamry, USA) was used as the basis for the integration of a metal-free anode water cycle and cathode exhaust water collector. Borosilicate glass bottles of 0.5 l to 1 l volume were used as the anode water cycle setup tank. A membrane pump (NF30 from KNF, Germany) and PE/PTFE tubes were used to feed the cell with water at a rate of ~300 ml min$^{-1}$. To maintain an elevated temperature of ~55–60 °C in the cell, the setup tank glass bottle is tempered by a heater plate (IKA, Germany). To initially clean the setup from ionic impurities a deionizer cartridge is used (Leycopure mixed bed resin from Leyco, Germany), which is bypassed during the actual dissolution experiment. The anode water samples were tapped from the cycle directly behind the cell. As there was no water cycling at the cathode side, the cathode water samples were taken from the PP cathode exhaust water collector bottle. Its reservoir volume was ~10 ml and was continuously flushed at the cathode water exhaust rate of 8.2 ± 0.4 ml h$^{-1}$.

*Measurement procedure.* Before starting the dissolution tests, the anode water cycle is cleaned up at elevated temperature for at least 12 h by running the water through the deionizer cartridge to remove eventual ionic impurities released from the setup (feed water processed by ULTRA CLEAR® TP ultrapure water system from Evoqua, USA). After the cleaning period, the cartridge is bypassed and the first 10 ml water sample (standard volume for all samples) is tapped from the cycle without contact to the cell as a clean reference. In the next step, the cell is mounted and water is run through the cell for 10 min without current. At the end, again 10 ml water is tapped from the cycle. Subsequently, the current density was set to 0.2 A cm$^{-2}$ for 10 min and another water sample was tapped. Thereafter, the current is set to 2 A cm$^{-2}$ where it stays for the rest of the experiment. Anode water samples are tapped after holding intervals of 10 min, 30 min, 1 h, 1 h, and successive every 24 h. Cathode samples are taken for the first time 3 h after the start of the test and then also every 24 h. During the aqueous tests with fixed anode side water volume the tapped water, the consumed water, and the water lost to the cathode by electroosmosis is replaced by ultrapure feed water. In contrast, during the acidic tests and the aqueous tests with variable water volume, the tapped, consumed or electroosmotically transported electrolyte volume is not replaced and the initial volume of water or 0.1 M $H_2SO_4$ is gradually reduced. At the end of the dissolution test, both flowfields were purged with nitrogen and the cell was held for at least 12 h at 1.2 V cell voltage to avoid anode side catalyst reduction by permeating hydrogen from the cathode side or the membrane. In addition, for the last two tests with gradually reduced water/acid volume, polarization curves and electrical impedance spectroscopy were made to reveal the end-of life performance of the aged MEAs. The MEAs are extracted in a glove bag and stored under nitrogen until spectroscopic analysis/STEM graphs were made.

### Scanning transmission electron microscopy
*Sample preparation.* Samples were embedded in Araldite 502 epoxy resin and cured overnight at 60 °C. Ultrathin sections with a nominal thickness of 100 nm were cut with an RMC Boeckeler PowerTome using a Diatome ultra 45° diamond knife. The sections were collected on copper grids for subsequent imaging via scanning transmission electron microscopy.

*Measurements.* STEM micrographs were taken with a Zeiss Crossbeam 540 FIB-SEM with annular STEM with a detector accelerating voltage of 20 kV and probe current of 300 pA. High-angle annular dark-field was chosen as imaging mode,

owing to its high contrast between atoms with low atomic number Z (membrane: F, C,..) and a high atomic number (Ir, Au, Ti). The chemical composition was determined via EDX (X-Max 150 silicon drift detector, Oxford Instruments; Software: Aztec Version 3.3, Oxford instruments) with an accelerating voltage of 20 pA and a probe current of 300 pA.

STEM micrographs were post-processed with ImageJ. As the samples have an average thickness of 100 nm while the penetrating depth of the electron beam can be several micrometers, EDX analysis contains background elemental information like copper from the TEM-grid. For reasons of simplification, only elements of interest (Pt, Ir, Au, Ti, C, F, and O) are shown in the elemental point measurements.

## Data availability

The authors declare that the main data supporting the findings of this study are available within the article and its Supplementary Information. Source data are provided with this paper. Extra data are available from the corresponding authors upon reasonable request. Source data are provided with this paper.

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

## Acknowledgements

J.K., M.M, M.R., and S.C. acknowledge funding by the German Federal Ministry of Education and Research (BMBF) within the Kopernikus Project P2X under the grant number 03SFK211-2. D.E.-L. and S.C. acknowledge financial support from Deutsche Forschungsgemeinschaft DFG under project number CH 1763/3-1 and 1763/4-1. We thank the chair of Technical Electrochemistry at the Technical University of Munich for providing the MEA preparation equipment. We thank Carina Schramm for electrode preparation.

## Author contributions
S.C. conceived and developed the idea and coordinated the work. J.K. and S.C. designed the experiments. J.K. prepared electrodes for aqueous measurements, performed dissolution measurements in aqueous electrolyte, liquid sample analysis, and data analysis. M.M. developed the GR-free MEA setup, prepared MEAs, performed MEA measurements, and data analysis. K.S. performed H-cell measurements. M.B. performed STEM measurements. T.B. performed ultramicrotome thin-film cuts for STEM. The manuscript was written by J.K., M.M., D. E.-L., and S.C. with input and feedback from all authors.

All authors contributed through scientific discussions and have given approval to the final version of the manuscript.

## Funding

## Competing interests
The authors declare no competing interests.
