## [Peer Review File · Nature Communications]

REVIEWER COMMENTS

Reviewer #1 (Remarks to the Author):

This is a very interesting approach to resolving a critical problem plaguing the electrolysis community, but applicable to various electrochemical conversion devices (CO₂ reduction in particular, alkaline electrolysis etc). That the environment in which we study the reactions at a "fundamental" level and the applied system are not the same. Similar to the pressure gap in heterogeneous catalysis. This manuscript is timely, relevant and in this reviewers opinion of high impact. I do have some questions and clarifications that are necessary but am encouraging of its publication.

Core issue:

-Disambiguating pH in AMS and pH in MEA, dissolution trends matching. Vs what's actually happening in an MEA and what is AMS. Line 218 I would make a separate paragraph and emphasize before transitioning. Its also good to move the timescale to after you discuss pH in MEA since that transition would be more natural.

-To this point, Fig 3 is all AMS data. You vary the pH in the solution, or you vary the catalyst loading or ionomer loading on the disk. The AMS needs to be emphasized. The takeaway here is that pH of the solution has the biggest impact on the S number.

- comparison of Nafion %, the way I read the trend here is that the baseline is at 33% Nafion and it bucks the trend, or at least its volcano like. Too much Nafion is bad, not enough is worse. To what extent would physical detachment of the Ir particles be accounted for by ICPMS? Would this lead to lower S factor (higher dissolution). This parameter is hard to understand why its causing this issue. The nafion should take up the solution and influence the local pH at catalyst embedded in "catalyst layer". Otherwise the different pHs (at 33% nafion) wouldn't readily show a difference either. If detachment of catalyst is an issue, this could explain the trend, otherwise it would be good to have a good explanation. Elaborate on your point... "Nafion plays a role in AMS measurements and must be tuned to reflect the dissolution pathway of interest". Something like that, but would be good to know why its all over the place (as much as why the bulk pH is increasing SF). I would offer this, its irrelevant. In the MEA we optimize the ionomer content for mass and charge transport, physical binding. Its a different parameter in the MEA. In your case you just need to have a binder for the catalyst. I would almost remove this parameter or put in SI or suggest it should be consistently applied but comparisons with different Nafion content can't be compared.

-line 321 then should be corrected. Yes there is a discrepancy between assumed pH and real pH of Nafion. Or at least its not as acidic as we estimate with AMS experiments. While local pH with MEA may be lower than "bulk" nafion pH, its still significantly higher than AMS pH. You could argue that in AMS the local pH is a bigger factor? Although based on your experiments, when local pH decreases, it would be reflected in the measurement as closer behavior to pH 1, which it isn't its distinct. "pH values under estimated" is a little ambiguous. Assumed to be low ~pH 1, and in reality probably higher. In accordance with the literature you site saying that it might be close to neutral.

Minor issues:

-line 294, systematic?

-Correct Fig 1 caption and line on left hand side.

-Fig 2 caption

-Fig 3 reviewers recommendation is to remove the white background to insets, would look more appealing

-Line 139 remove reference to chapter

-Nafion content with addition of sulfuric acid always?

-"Mass transport conditions were excluded as explanation by varying layer thickness and flow rate. Thus, other reasons have to be responsible for this behavior."

-why is there an increase in Ir dissolution for pH 3 at the end?

-re catalyst loading and dissolution. I think here you need to be maybe more specific, that there is no influence wrt to AMS measurements. However that in an MEA with conventional PTL interface with CL.

The catalyst loading reduction may play a role in Ir dissolution and stability. The way it reads right now it says that it doesn't matter. Reviewer thinks this is not yet known in full MEA setups.

-why does voltage increase at fixed current with increasing catalyst loading. Usually in MEA this is either constant or improved performance at higher loading. Note this is related to point above. While the dissolution trend is also a bit harder to interpret. (Fig 4 supplemental).

-similarly with changing pH, pH3 the voltage is the worse (performing) but the dissolution is lower. This also seems counterintuitive. I guess the S number is nice for comparison. But there are some details here that are relevant and interesting to daylight.

-fig 16, why does the iR corrected cell voltage get worse with acid compared to water. This is also counterintuitive.

-ref 44 needs an author

-for fig 4. why does S number improve for MEA and stabilize for AMS, while the Ir dissolution trend stabilizes and increases, respectively. At a fixed current, this is counterintuitive.

-Alia et al papers should be referenced and commented on. There are comparisons there for Ir dissolution in AMS and MEA as well. They were looking for general trends vs how to make them match.

Reviewer #2 (Remarks to the Author):

This paper addresses the discrepancy between anode catalyst dissolution during oxygen evolution reaction (OER) in aqueous model systems and under membrane electrode assemblies (MEA) operation.

This aspect is of great relevance for PEM electrolysis that is one of the most promising technologies for producing green hydrogen.

In this regard, this paper is of interest for the hydrogen scientific community.

However, some aspects need to be carefully considered.

1. Selection of H₂SO₄ as the liquid electrolyte in the experiments carried out here in half cell seems to be not very appropriate for addressing these aspects. The liquid electrolyte analogous of perfluorosulphonate membrane is trifluoromethanesulfonic acid [see for example Langmuir 1986, 2, 4, 393–405; J. Electrochem. Soc. 1989, 136, 3369]. A careful comparison would require using this electrolyte. Alternatively, perchloric acid is more appropriate than sulphuric acid since the corresponding anion does not adsorb on the catalyst surface at the same extent of sulphate anions. This of course generate significant discrepancy.

2. The other aspect that needs to be addressed is the effect of the different catalyst loading typically used in half-cell studies and in MEAs. A correlation between the degradation rate in PEM electrolysis and the anode catalyst turnover frequency has been identified in the literature.

This aspect should be properly discussed. Specifically, the degradation rate increases with a reduction of the anode catalyst loading whereas catalyst utilisation can be different in the two systems especially in consideration of the used electrolyte.

3. Local pH effects are also significantly different depending on the used electrolyte. This aspect is of large relevance and should be carefully considered. As an example, besides varying the Nafion ionomer content in the electrodes, it would be also useful using ionomers with different ion exchange capacity.

4. Recirculation of water at high stoichiometry through the anode in single cell/stack studies is another relevant aspect that differentiate the behaviour of PEMWEs from half-cell operation. This has an impact on the anode catalyst behaviour.

5. Typical MEA durability studies for water electrolysis indicate that the major losses, both recoverable and unrecoverable, occur in the first 100-200 h. The long-term MEA behaviour is different from these first hundred hours often assumed as conditioning period. The time scale in Fig. 4 regards a minor interval. This aspect needs also proper consideration.

6. There are no clear conclusions from this work beside the fact that a discrepancy is evident. The

discrepancy between half-cell and MEAs studies was already well known for PEM water electrolysis. Thus, which are the specific indications to overcome these issues that can be derived from this work. How, half-cell studies should be carried out to get proper information that can be of interest for practical operation of these catalysts in MEAs? These aspects should be in my opinion specifically addressed.

Reviewer #3 (Remarks to the Author):

In this work, Knoppel and coworkers developed a cell to probe the difference existing between OER catalyst stability measured in aqueous tests and that measured in MEA configuration. This works represent a major step toward achieving a better understanding of these systems, and the unique setup and testing protocols are very carefully executed. However, while I would like to praise the authors for their careful execution, several critical questions remain unanswered after reading this manuscript, and sometimes explanation brought in by the authors are not backed up by any measurements. This is especially the case regarding the pH of Nafion, which is brought in to explain the difference in stability for the Ir-catalyst, but is never measured or no additional tests were performed to conclusively discard other possible explanations. Thus, I believe that this work can potentially be published in Nature Communications, but only after answering these questions with additional measurements.

Comments:

Regarding the effect of Nafion loading, the authors mention that tests were made varying the thickness of the catalyst layer as well as flow rate. However, no such data can be found in the SI. Can the authors present these data for the readers?

Furthermore, would it be possible to intentionally trap water containing dissolved Iridium into porous Nafion layers with different thicknesses before to flush "fresh" DI water on one side and collect by ICP-MS on the other side the water coming out, to simply observe that the morphology of the Nafion layer largely affects the retention time for dissolved iridium species?

Regarding the effect of Nafion loading on the S-number, worth mentioning that similar explanation regarding local saturation were given by the group of Markovic to explain the better stability of nanoporous catalysts, or even by the group of Nocera/Costentin to study the kinetics of "self-healing" for Co-Pi OER catalysts. These papers might be cited as a reference for reader to understand better how such saturation will affect the kinetics for redeposition (especially the kinetics analysis by Costentin).

As explained by the authors, the fact that the Nafion loading is smaller by a factor of 3-4 in the MEA when compared to the baseline measurements would decrease the lifetime of the Ir-based catalysts in the MEA configuration, which is not the case. Then, what explains such discrepancies between these two observations? It would be critical to explain this, as without such explanation this work, even though very carefully done, fails at providing any guidelines for researchers to optimize the MEA configuration (or even the catalysts preparation in RDE configuration) to increase the catalyst's lifetime.

Regarding the effect of pH, again this work while very beautiful could be improved by providing more physical insights to explain the observed dissolution rates. Hence, could one envision using a reference electrode (or simply a Pt electrode in a H₂-saturated atmosphere) embedded into the Nafion layer in order to estimate its local pH? It appears rather unsatisfactory to simply state that more work would be needed to conclude that indeed Nafion operates at a pH close to 1 as a way to rationalize that the dissolution rate observed in the MEA configuration is closed to the one observed at pH 1.5. Furthermore, as discussed above, stabilization (or self-healing) of catalysts can be achieved through redeposition of dissolved Ir-species on the surface of the catalyst, which kinetics is dependent on the local concentration of dissolved Ir-species and thus on the volume of electrolyte. Hence, rather than the effect of current on the crystallization of the Ir catalysts, which is highly debatable, different measurements using similar H-cell but different volume of electrolytes could be carried out to understand if such self-healing process is at play for Ir-based catalysts.

Figure 5a and 5b appears rather unnecessary/non-informative, especially since the color code is not respected for DI water (dark blue in Figure 5a and black in Figure 5c).

The observation of Au particles formed at the interface between the membrane and the catalyst layer should raise questions regarding the stability itself of the Au-coated Ti flow plates under flow of acidic solutions. With this in mind, the major issue regarding this work comes from the fact that one can question if the effect of pH observed in Figure 3f is simply not coming from a difference of stability of the PTL at different pH.

Finally, the main critics regarding the S-number calculation relies on the way the current density is taken into account. Indeed, the authors should specify that the amount of oxygen evolved as defined in the S-factor is not directly measured (unlike for the amount of dissolved cation), but rather estimated from the anodic current. Hence, even though the effect of current density is in part retrieved by accounting for it in the S-number calculation, it is not completely suppressed as different current density will mean different local pH as well as difference equilibrium at the interface which can certainly destabilize the catalyst. This could also modifies the stability of the catalyst. In order to suppress such uncertainty, measurements at fixed current density (rather than fixed applied potential) can be carried out using different pH, to disentangle both effects.

Side comments:

What creates such noise in the online ICP-MS measurements when a large amount of dissolved iridium is added?

Unlike stated in the manuscript, Nafion is in fact needed for OER measurements in three electrodes RDE systems testing Ir oxide catalysts in the powder form to ensure the mechanical integrity of the thin catalyst layer upon rotation (while I agree that such Nafion is not needed if the catalyst was prepared by electrodeposition).

Rebuttal letter Nature Communications

Ms. Ref. No.: NCOMMS-20-49431

On the limitations in assessing stability of oxygen evolution catalysts using aqueous model electrochemical cells.

REVIEWER COMMENTS

We thank the editor and reviewers for their time and consideration. Below, we have addressed all comments and questions point by point. The sentences added in the main text are highlighted in yellow.

Reviewer #1 (Remarks to the Author):

This is a very interesting approach to resolving a critical problem plaguing the electrolysis community, but applicable to various electrochemical conversion devices (CO₂ reduction in particular, alkaline electrolysis etc). That the environment in which we study the reactions at a "fundamental" level and the applied system are not the same. Similar to the pressure gap in heterogeneous catalysis. This manuscript is timely, relevant and in this reviewers opinion of high impact. I do have some questions and clarifications that are necessary but am encouraging of its publication.

We thank the reviewer for the overall positive evaluation of our manuscript and her/his assessment that our work is timely, relevant, and in her/his opinion, of high impact. The answers to the reviewer's questions are shown below in blue.

Core issue:

-Disambiguating pH in AMS and pH in MEA, dissolution trends matching. Vs what's actually happening in an MEA and what is AMS. Line 218 I would make a separate paragraph and emphasize before transitioning. Its also good to move the timescale to after you discuss pH in MEA since that transition would be more natural.

We thank the reviewer for this comment. As suggested by the reviewer, we have inserted a separate paragraph to fully emphasize the point raised in line 218. To further emphasize the transition from AMS to MEA more clearly in the manuscript, we have changed the wording of the paragraph from

"While Nafion in the literature is assumed to operate at highly acidic conditions [1, 2], these results indicate that pH in anode catalyst layers of MEA might have been underestimated in previous studies. Further experiments in MEA systems have to be conducted to unravel the magnitude of the differences."

To

"MEA environments for PEMWE are, according to the literature, highly acidic due to the use of Nafion [1, 2]. The observations, made here in AMS, however, indicate, that the activity of protons in MEA electrolysis might be lower than generally accepted. Further experiments in MEA systems have to be conducted to unravel the magnitude of the differences. "

Furthermore, we shifted the discussion of pH in the electrolyte upfront of the discussion of timescale in both systems according to the reviewer's suggestions.

-To this point, Fig 3 is all AMS data. You vary the pH in the solution, or you vary the catalyst loading or ionomer loading on the disk. The AMS needs to be emphasized. The takeaway here is that pH of the solution has the biggest impact on the S number.

We thank the reviewer for pointing out that our caption is not detailed enough. We have added a phrase to the main caption to clarify the measurement method. The caption reads now: "**Figure 3 S-numbers of IrO_x catalyst spots with varied conditions, measured in SFC-ICP-MS. a)[...]**".

- comparison of Nafion %, the way I read the trend here is that the baseline is at 33% Nafion and it bucks the trend, or at least its volcano like. Too much Nafion is bad, not enough is worse. To what extent would physical detachment of the Ir particles be accounted for by ICPMS? Would this lead to lower S factor (higher dissolution). This parameter is hard to understand why its causing this issue. The nafion should take up the solution and influence the local pH at catalyst embedded in "catalyst layer". Otherwise the different pHs (at 33% nafion) wouldn't readily show a difference either. If detachment of catalyst is an issue, this could explain the trend, otherwise it would be good to have a good explanation.

Avoiding particle detachment is rather an issue of instrument maintenance than a concern during actual SFC-ICP-MS data acquisition. If catalyst particles detach within a measurement, the most significant threat is that the nebulizer, which disperses the samples for further injection into the ICP-MS spray chamber, is ultimately clogged. If clogging is severe, this would prevent any sample introduction to the ICP-MS, so SFC-ICP-MS measurements would not be feasible.

However, if catalyst particles detach and pass the nebulizer, they can be easily identified in the ICP-MS data. If a particle (instead of dissolved material) enters the plasma and is ionized, a spike at a single data point would appear, with much higher counts than the surrounding ICP-MS data acquired. We did not see such spikes in the data for 5% Nafion. Therefore, we can exclude particle detachment as source for the lower Ir stability observed.

Elaborate on your point... "Nafion plays a role in AMS measurements and must be tuned to reflect the dissolution pathway of interest". Something like that, but would be good to know why its all over the place (as much as why the bulk pH is increasing SF). I would offer this, its irrelevant. In the MEA we optimize the ionomer content for mass and charge transport, physical binding. Its a different parameter in the MEA. In your case you just need to have a binder for the catalyst. I would almost remove this parameter or put in SI or suggest it should be consistently applied but comparisons with different Nafion content can't be compared.

We agree with the reviewer that the role of Nafion, and its relative content in the catalyst layer, are clearly different in AMS and MEA systems and that the one-to-one comparison is impossible. Nevertheless, since Nafion is used in both systems, its effect on Ir stability is worth a consideration. Nafion loadings have been found to diverge significantly, not only when comparing AMS-MEA systems but also across research studies performed in AMS. Indeed, catalyst layers prepared for previous AMS studies in the literature presented Nafion contents ranging from 0% [3] to 33% [4]. Thus, it is a factor that merited consideration. Hence, as a means transparency toward the electrocatalysis community, we advocated to

perform such Nafion content study and we believe that shifting it to the supporting information would weaken the information gathered from it. Thus, even if inconclusive, we would opt to keep it in the main manuscript so that the interested reader can independently assess the implications of these measurement.

-line 321 then should be corrected. Yes there is a discrepancy between assumed pH and real pH of Nafion. Or at least its not as acidic as we estimate with AMS experiments. While local pH with MEA may be lower than "bulk" nafion pH, its still significantly higher than AMS pH. You could argue that in AMS the local pH is a bigger factor? Although based on your experiments, when local pH decreases, it would be reflected in the measurement as closer behavior to pH 1, which it isn't its distinct. "pH values under estimated" is a little ambiguous. Assumed to be low ~pH 1, and in reality probably higher. In accordance with the literature you site saying that it might be close to neutral.

The reviewer raised a valid point about the phrasing of our conclusions. We have changed the wording in line 321 and added a separate sentence.

The former sentence was

"The results shown in this study indicate that pH values in the literature have been gravely underestimated so far."

We changed it to:

"The results shown in this study indicate that proton activity in MEA environment is supposedly lower as estimated from the concentrations. Thus, it is likely, that the effective pH in MEA environment is less acidic as generally accepted in the literature. "

Minor issues:

-line 294, systematic?

-Correct Fig 1 caption and line on left hand side.

The unwanted line at the left-hand side of figure 1 was removed. Furthermore, the caption

"[...] **a)** Schematic drawing of degradation processes in a classical aqueous three-electrode cell in aqueous electrolyte [...]"

Was changed to

"[...] **a)** Schematic drawing of degradation processes in a classical aqueous electrolysis cell in aqueous electrolyte [...]"

-Fig 2 caption

The caption of figure 2 was modified to emphasize the content more clearly.

"**Figure 1 Dissolution profiles of IrO_x catalyst spots in SFC-ICP-MS measurements.** **a)** Applied current step and **b), c)** resulting dissolution of **b)** a baseline measurement at standard conditions (10 μg_{Ir} cm⁻² catalyst loading, 200 μl min⁻¹ flow rate, fresh electrolyte, 33 wt% Nafion in the catalyst layer and 0.1 M H₂SO₄ (pH =1)) and **c)** with a variation of electrolyte pH. The integration area for calculation of S-numbers is highlighted by vertical dashed lines. "

-Fig 3 reviewers recommendation is to remove the white background to insets, would look more appealing

We appreciate the reviewers' comment, but captions would be hardly readable given the amount of colours scheme used. Prior to settling on the white inset background scheme, many combinations were attempted, and the one with white background provided the best contrast to ensure all insets were visible without the white insets.

-Line 139 remove reference to chapter

The chapter reference has been removed following the suggestion by the reviewer.

-Nafion content with addition of sulfuric acid always?

We would like the reviewer to specify and/or rephrase the comment raised, phrasing is not fully understandable in its current state.

-"Mass transport conditions were excluded as explanation by varying layer thickness and flow rate. Thus, other reasons have to be responsible for this behavior."

For ease of clarity we have removed the sentence from the main manuscript.

-why is there an increase in Ir dissolution for pH 3 at the end?

The question raised by the reviewer is very interesting. Two processes are known to mostly increase dissolution of iridium: OER and transitions in the oxidation state. At the shutdown, of the constant current mode, the potential decreases. Thus, transient dissolution, through a change in the oxidation state of the iridium based catalyst is likely. If the reviewer compares closely the other dissolution profiles during constant current measurements, she/he will be able to see a slight increase upon shutdown of OER as well, but not at the same extent as seen in pH 3. A follow-up study, of dissolution dependencies from pH 0 to 14, which goes beyond the scope of this work, will clarify the issue.

-re catalyst loading and dissolution. I think here you need to be maybe more specific, that there is no influence wrt to AMS measurements. However that in an MEA with conventional PTL interface with CL. The catalyst loading reduction may play a role in Ir dissolution and stability. The way it reads right now it says that it doesn't matter. Reviewer thinks this is not yet known in full MEA setups.

We agree with the reviewer that this difference was not been clearly pointed out in the first version of the manuscript. Thus, we expanded the relevant sentence in line 292 to make the distinction more evident.

"These differences cannot be related to a sole factor. Loading, flow rate, dissolved species, timescale or Nafion content, although not relevant factors in AMS, might play a role in MEA."

-why does voltage increase at fixed current with increasing catalyst loading. Usually in MEA this is either constant or improved performance at higher loading. Note this is related to point above. While the dissolution trend is also a bit harder to interpret. (Fig 4 supplemental).

We would like to point out to the reviewer that our current density in SFC-ICP-MS is normalized to loading of catalyst spots. Thus, the absolute current density increases at higher loadings. In our SFC, the microfluidic channels induce an ohmic drop of $\sim 150 \Omega$. Furthermore, the mass transport in the catalyst layer is hindered at higher loadings in the SFC. Bubble transport efficiency from the surface decreases at higher current densities. Thus, a higher ohmic drop and mass transport limitations are responsible for increased loadings' increased potential.

The captions of supplementary figures 4-9 were extended to emphasize the experimental conditions

-similarly with changing pH, pH3 the voltage is the worse (performing) but the dissolution is lower. This also seems counterintuitive. I guess the S number is nice for comparison. But there are some details here that are relevant and interesting to daylight.

The answer to this question is similar to the previous answer. Although the ionic conductivity is enhanced with K_2SO_4 salt at higher pH in experiments with different pH, it is still lower than in the 0.1 M electrolyte in the baseline measurement, as sulfate and potassium ions have lower ionic mobility than the protons being substituted in the electrolyte. Thus, a higher ohmic drop is observed in electrolytes with higher pH.

Furthermore, we would like to point out that, even if it is counterintuitive, iridium dissolution in aqueous electrolyte at otherwise constant conditions is only dependent on the transferred charge, as long as a potential of $E_{RHE}=1.8 \text{ V}$ is not surpassed (this would trigger additional dissolution processes [5]). The supplementary information of the original S-number publication features a series of experiments at different current densities (and thus, different potentials), and the only factor correlating to iridium dissolution was the amount of transferred charge [4]. As the potentials in our constant current measurements are below this threshold, our measurements indeed show increasing stability of iridium as OER catalysts at higher pH.

-fig 16, why does the iR corrected cell voltage get worse with acid compared to water. This is also counterintuitive.

We assume, that the reviewer refers to supplementary figure 16. The data shown here in the text is end of life data for both MEAs. As shown in the STEM-micrographs, the acidic MEA is almost fully degraded at the end of life. Through the electrolytic contact to the PTL and bipolar plates, OER can also directly take place on these. However, it is likely that the onset of OER first occurs on the remaining anode catalyst layer, as iridium is the most catalytically active.

We do not argue here with the full potential curve, but only onset potential and it is only one hint. To clarify the issue, we changed the caption to supplementary figure 16 to

"a) iR -drop corrected **end of life** polarization curves in the precipitation free MEA setup with anode catalyst loadings of $1 \mu\text{g}_{\text{Ir}} \text{ cm}^{-2}$ with DI water and 0.1 M H_2SO_4 in the anode water cycle"

-ref 44 needs an author

We apologize for the missing author in reference 44. We modified our library accordingly.

-for fig 4. why does S number improve for MEA and stabilize for AMS, while the Ir dissolution trend stabilizes and increases, respectively. At a fixed current, this is counterintuitive.

The amount of dissolved material shown in figure 4 refers to the total Ir dissolved. Ir dissolution rates, in contrast, are calculated with the relations discussed in supplementary Note 2. As the rate only depends on the dissolved amount in the previous time interval ($n_{Ir}(t_i) - n_{Ir}(t_{i-1})$), the S-number does not correlate linearly to the total dissolved amount.

If the total dissolved amount grows linearly (constant dissolution rate), the S-number is mostly constant (AMS). If the total dissolved amount of iridium increases non-linearly and shows nearly asymptotic behavior, the iridium dissolution decreases and thus, the S-number increases.

-Alia et al papers should be referenced and commented on. There are comparisons there for Ir dissolution in AMS and MEA as well. They were looking for general trends vs how to make them match.

The authors thank the reviewer for pointing out additional literature strengthening our initial work hypothesis. We changed the sentence

"However, all evaluations of catalyst stability in aqueous systems do not represent the conditions in PEMWE."

In the introduction to

"However, comparative data of catalyst stability in both systems show that degradation in aqueous systems does not represent the conditions in PEMWE [6]"

To account for the mentioned works.

Reviewer #2 (Remarks to the Author):

This paper addresses the discrepancy between anode catalyst dissolution during oxygen evolution reaction (OER) in aqueous model systems and under membrane electrode assemblies (MEA) operation.

This aspect is of great relevance for PEM electrolysis that is one of the most promising technologies for producing green hydrogen.

In this regard, this paper is of interest for the hydrogen scientific community. However, some aspects need to be carefully considered.

We thank the reviewer for taking the time to review our manuscript and his assessment that our work is of great relevance for PEM electrolysis.

1. Selection of H₂SO₄ as the liquid electrolyte in the experiments carried out here in half cell seems to be not very appropriate for addressing these aspects. The liquid electrolyte analogous of perfluorosulphonate membrane is trifluoromethanesulfonic acid [see for example Langmuir 1986, 2, 4, 393–405; J. Electrochem. Soc. 1989, 136, 3369]. A careful comparison would require using this electrolyte. Alternatively, perchloric acid is more appropriate than sulphuric acid since the corresponding anion does not adsorb on the catalyst surface at the same extent of sulphate anions. This of course generate significant discrepancy.

We agree with the reviewer that the choice of H₂SO₄ in hindsight might not be optimal. However, the use of H₂SO₄ is still widespread in fundamental research of OER catalysts. In comparison to HClO₄, an electrolyte recommended by the reviewer, we only foresee marginal differences in stability compared to H₂SO₄ used in this study. Indeed, 0.1 M HClO₄ has been used to study stability of the same Alfa Aesar IrO_x catalyst in an earlier study to which our group contributed [7]. The S-numbers of 3×10^4 shown there

resemble the S-numbers of the AMS baseline measurements of 6×10^4 shown in this study. The difference can be led back to a slightly different calculation method (steady state calculation vs full chronopotentiometric hold including peaks at the beginning and the end). Furthermore, HClO_4 was used in the original paper, where the S-number was introduced [4]. Here, the S-numbers of various IrO_x based catalytic materials lie in the same order of magnitude as our AMS baseline measurement. Therefore, we conclude that the choice of acidic electrolyte, although interesting, only yields a minor impact on the results shown.

2. The other aspect that needs to be addressed is the effect of the different catalyst loading typically used in half-cell studies and in MEAs. A correlation between the degradation rate in PEM electrolysis and the anode catalyst turnover frequency has been identified in the literature.

This aspect should be properly discussed. Specifically, the degradation rate increases with a reduction of the anode catalyst loading whereas catalyst utilisation can be different in the two systems especially in consideration of the used electrolyte.

The question raised by the reviewer is of great relevance for the practical application of PEM water electrolysis. However, to perform a study with varying loadings in MEA system is not a straightforward task. Loading directly impacts parameters such as catalyst layer thickness, contact resistance and mass transport in the MEA. Thus, for ease of simplification (i.e. ruling out these factors in our AMS-MEA stability discrepancy) we focused on a specific catalyst loading.

The issues raised by the reviewer are then beyond the scope of this manuscript, and will be addressed in a separate study in the future.

3. Local pH effects are also significantly different depending on the used electrolyte. This aspect is of large relevance and should be carefully considered. As an example, besides varying the Nafion ionomer content in the electrodes, it would be also useful using ionomers with different ion exchange capacity.

We agree with the reviewer that the polymer electrolyte choice is likely to affect the local pH in the anode catalyst layer of the MEA. However, Nafion is the current standard ionomer for PEM water electrolysis up to this day for large scale industrial application. In line with the previous suggestion by the reviewer, studying local pH effects of different polymer electrolytes is a promising approach, but it goes beyond the scope of this manuscript.

4. Recirculation of water at high stoichiometry through the anode in single cell/stack studies is another relevant aspect that differentiate the behaviour of PEMWEs from half-cell operation. This has an impact on the anode catalyst behaviour.

We are not sure what the reviewer means with "Recirculation of water at high stoichiometry". We believe that the reviewer is referring to either the difference in water flow rates in AMS and MEA or the recirculation of dissolution products in the system.

With regards to the flow rate, both AMS (SFC-ICP-MS) and MEA are subjected to a flow over the catalyst layer. Naturally, the mass transport in both systems diverges. However, flow rate has been shown to have a negligible effect on iridium dissolution in AMS flow rate experiments. Therefore, it is questionable to which extent it should affect dissolution in MEA differently.

We hope that this answers the concerns raised by the reviewer.

5. Typical MEA durability studies for water electrolysis indicate that the major losses, both recoverable and unrecoverable, occur in the first 100-200 h. The long-term MEA behaviour is different from these first hundred hours often assumed as conditioning period. The time scale in Fig. 4 regards a minor interval. This aspect needs also proper consideration.

We agree with the reviewer that the most changes in the MEA occur after a timescale of 100-200 hours. Before this working time, changes still occur in MEA. Thus, the baseline of our MEA measurements, used in figures 1 and 6, is calculated from a measurement of ~250 h. For the full data, we would like to refer the reviewer to supplementary figure 2.

The extract of the first 48 h in MEA, shown in figure 4 in the main text, is merely for comparing two systems. Although the first 48 h are not fully representative of the MEA, we see most of the changes in stability occurring in this timeframe.

6. There are no clear conclusions from this work beside the fact that a discrepancy is evident. The discrepancy between half-cell and MEAs studies was already well known for PEM water electrolysis. Thus, which are the specific indications to overcome these issues that can be derived from this work. How, half-cell studies should be carried out to get proper information that can be of interest for practical operation of these catalysts in MEAs? These aspects should be in my opinion specifically addressed.

Although a discrepancy was already evident, this manuscript is the first report of a systematic assessment of such discrepancy. In the manuscript, we have already envisioned the option of utilizing gas diffusion electrodes (GDE) for electrochemical testing in half cell studies. These electrodes would realistic current densities and spatial separation between reactant, anode catalyst layer and electrolyte.

However, we acknowledge that this option has not been fully emphasized in our work's conclusions and outlook. Therefore, we added the following sentence in our concluding remarks.

"For this purpose, GDE systems currently under development might help to study OER catalysts in model systems which resemble the conditions in MEA much better."

Reviewer #3 (Remarks to the Author):

In this work, Knoppel and coworkers developed a cell to probe the difference existing between OER catalyst stability measured in aqueous tests and that measured in MEA configuration. This works represent a major step toward achieving a better understanding of these systems, and the unique setup and testing protocols are very carefully executed. However, while I would like to praise the authors for their careful execution, several critical questions remain unanswered after reading this manuscript, and sometimes explanation brought in by the authors are not backed up by any measurements. This is especially the case regarding the pH of Nafion, which is brought in to explain the difference in stability for the Ir-catalyst, but is never measured or no additional tests were performed to conclusively discard other possible explanations. Thus, I believe that this work can potentially be published in Nature Communications, but only after answering these questions with additional measurements.

We thank the reviewer for taking the time to review our work and his positive feedback regarding the execution of our study.

Comments:

Regarding the effect of Nafion loading, the authors mention that tests were made varying the thickness of the catalyst layer as well as flow rate. However, no such data can be found in the SI. Can the authors present these data for the readers?

We are sorry that the reviewer could not clearly identify our data regarding the variation of flow rate and loading variation. Such data was shown in the supplementary information in supplementary figures 4 and 5, to which we gratefully refer the reviewer. We hope that the reviewer can successfully identify the data.

Furthermore, would it be possible to intentionally trap water containing dissolved Iridium into porous Nafion layers with different thicknesses before to flush "fresh" DI water on one side and collect by ICP-MS on the other side the water coming out, to simply observe that the morphology of the Nafion layer largely affects the retention time for dissolved iridium species?

We thank the reviewer for this suggestion. Studies like the suggested one are in principle feasible. A different study design was indeed carried out for Pt as a fuel cell catalyst recently [8]. Similarly to Pt, Ir forms cations during dissolution. In the cited publication, the diffusion coefficient of Pt in Nafion was determined to be one order of magnitude lower than the diffusion coefficient in water.

The platinum results show a different retention time with Nafion than without Nafion. However, the membrane thickness used in the Pt catalyst layer thickness study was significantly higher compared to Nafion layers in a catalyst layer. While our catalyst spots for SFC-ICP-MS measurements are significantly below 1 μm , Nafion layer thickness starts at 5 μm . A much higher Nafion content would be needed to show significant retention time. Our group's unpublished results with iridium thin films have shown comparable behavior, as shown with comparable raw data in the figure below.

Figure 1 Delay times of dissolution peaks as a function of Nafion layer thickness measured in SFC-ICP-MS

With these results, we are confident that the Ir retention time within Nafion does not significantly affect our study's outcome significantly, as our experimental design always uses data from a quasi-steady state to calculate the S-number descriptor for stability.

Regarding the effect of Nafion loading on the S-number, worth mentioning that similar explanation

regarding local saturation were given by the group of Markovic to explain the better stability of nanoporous catalysts, or even by the group of Nocera/Costentin to study the kinetics of "self-healing" for Co-Pi OER catalysts. These papers might be cited as a reference for reader to understand better how such saturation will affect the kinetics for redeposition (especially the kinetics analysis by Costentin).

We appreciate the insights given by the reviewer. The authors of this manuscript are only aware of one dissolution/precipitation equilibrium for iridium, shown by Alexis Grimauds group [9]. In this work, the authors chemically dissolved Ir(V) based perovskites in HClO₄ and performed cyclic voltammetry with the resulting Ir-containing electrolyte on a freshly polished glassy carbon plate, resulting in a deposition of IrO_x on the glassy carbon surface.

We repeated the cycling with electrochemically dissolved iridium containing electrolyte, which was used for the experiments with dissolved ions in the electrolyte (supplementary figure 6). The iridium concentration was 84 μg l⁻¹. The results of the first and the 30th CV are shown in the figure below.

Figure 2 Cyclic voltammetry of a glassy carbon electrode in 0.1 M H₂SO₄ with 84 μg l⁻¹ electrochemically dissolved iridium.

Unlike the results shown Grimaud et al. for chemically dissolved perovskites, we do not see the formation of IrO_x features with electrochemically dissolved iridium. Hence, combined with the results shown in the manuscript, we do not believe there is a dissolution/precipitation equilibrium for electrochemically dissolved iridium in OER.

As explained by the authors, the fact that the Nafion loading is smaller by a factor of 3-4 in the MEA when compared to the baseline measurements would decrease the lifetime of the Ir-based catalysts in the MEA configuration, which is not the case. Then, what explains such discrepancies between these two observations? It would be critical to explain this, as without such explanation this work, even though very carefully done, fails at providing any guidelines for researchers to optimize the MEA configuration (or even the catalysts preparation in RDE configuration) to increase the catalyst's lifetime.

We agree with the reviewer that a lowered Nafion content in the anode catalyst layer of the MEA might negatively impact the stability of iridium in the MEA. Thus, a comparison between both systems with similar Nafion content would be desirable. However, different constraints have to be met for measurements in both systems. In SFC-ICP-MS, Nafion acts predominantly as a binder in the catalyst layer. The constraint is here the maintenance of the ICP-MS device, as particle detachment from the catalyst layer can damage the nebulizer.

On the other hand, in the more complex MEA system, Nafion has not only influence on the integrity of the catalyst layer, but also on features such as the contact resistance with the PTL. Optimizations in MEA have shown an optimal Nafion content of the anode catalyst layer around 10% [10].

We anticipated that this different Nafion contents might already cause some variations in the data trends. Indeed, this is the reason why such factor is mentioned explicitly. However, to study the effects of Nafion content on stability would need an optimization study on stability as a function of Nafion content in the MEA, which goes beyond this manuscript's scope. A further study will elaborate on this topic.

However, we see shortcomings in our discussion regarding Nafion content. Thus, parts of the answer to the reviewer were added to L 209 in the manuscript.

"[...]different constraints have to be met for measurements in both systems. In SFC-ICP-MS, Nafion predominately acts as a binder in the catalyst layer to avoid particle detachment. On the other hand, in the more complex MEA system, Nafion has not only influence on the integrity of the catalyst layer, but also on features such as contact resistance with the PTL. As[...]"

Regarding the effect of pH, again this work while very beautiful could be improved by providing more physical insights to explain the observed dissolution rates. Hence, could one envision using a reference electrode (or simply a Pt electrode in a H₂-saturated atmosphere) embedded into the Nafion layer in order to estimate its local pH? It appears rather unsatisfactory to simply state that more work would be needed to conclude that indeed Nafion operates at a pH close to 1 as a way to rationalize that the dissolution rate observed in the MEA configuration is closed to the one observed at pH 1.5.

The introduction of a reference electrode in MEA is a very painstaking process. This has been achieved in the past by Brightman et al. [11]. The authors show that the MEAs cathode's potential in an MEA drops linearly at increasing current density. At constant current density, as in the comparison between DI water and 0.1 M H₂SO₄, the cathode side serves as a pseudo reference electrode. At constant current, both the rate at which H₂ is produced its H₂ pressure are constant.

The measurement of pH in MEA, on the other hand, is extremely challenging. First, it is debatable if the concept of pH is even fully transferrable from aqueous systems to polymer electrolytes. For example, the proton concentration in polymer electrolyte has been shown to resemble the proton concentration of a strong acid. However, through water swelling up in the membrane and protons conduct along bonds of the Nafion membrane, the protons' effective activity might be lower.

As shown in supplementary note 8 and supplementary figure 16, the onset potential of the acidified MEA is around 120 mV higher compared to the conventional MEA. Although this is end-of-life data, and the acidified MEA has suffered supposedly higher degradation in MEA, it is foreseeable that the OER onset starts first at the system's material with the lowest overpotential for OER, which is Ir at the anode catalyst layer. This observation is complimentary to observations in the AMS (Figure 3 f), Supplementary figure 6), and leads to the conclusion, that effective pH in MEA is higher than expected.

Although we acknowledge the importance of such suggested studies, a dedicated setup (with its shortcomings) should be purposefully devised to evaluate local pH effects, beyond the scope of the core message of this study: to showcase the stability discrepancy of OER catalysts in AMS vs MEA systems.

Furthermore, as discussed above, stabilization (or self-healing) of catalysts can be achieved through redeposition of dissolved Ir-species on the surface of the catalyst, which kinetics is dependent on the local concentration of dissolved Ir-species and thus on the volume of electrolyte. Hence, rather than the effect

of current on the crystallization of the Ir catalysts, which is highly debatable, different measurements using similar H-cell but different volume of electrolytes could be carried out to understand if such self-healing process is at play for Ir-based catalysts.

As discussed above, we are fairly confident that no redeposition occurs in AMS due to the data presented in supplementary figure 6 and the unpublished data shown above.

Figure 5a and 5b appears rather unnecessary/non-informative, especially since the color code is not respected for DI water (dark blue in Figure 5a and black in Figure 5c).

We would like to point out to the reviewer that we are unaware of such a non-respected color code in figure 5. Figure 5 a) shows DI water in light blue, Figure 5 b) shows the same dark blue as used for the bar diagram.

However, the authors acknowledge that the color code used in schemes is non-consistent with the color code, used in Figures 1 a) and Figure 6. We changed the color code in this figures accordingly, so it is consistent across the whole manuscript.

The observation of Au particles formed at the interface between the membrane and the catalyst layer should raise questions regarding the stability itself of the Au-coated Ti flow plates under flow of acidic solutions. With this in mind, the major issue regarding this work comes from the fact that one can question if the effect of pH observed in Figure 3f is simply not coming from a difference of stability of the PTL at different pH.

We understand the concerns raised by the reviewer regarding the inherent stability of Au-coated Ti PTLs at different electrolyte environments. These can be clarified as follows:

1. Figure 3 consists fully of SFC-ICP-MS (AMS) measurements. Thus, the diverging S-numbers at different pH values shown in this figure cannot be explained by different MEA components' stability. We extended the caption of figure 3, also requested by reviewer 1. The caption is now **"Figure 3 S-numbers of IrO_x catalyst spots with varied conditions, measured in SFC-ICP-MS. a) [...]"**.
2. The STEM micrographs shown in the supplementary information are taken after a full 48 h of measurements. However, the S-numbers in figure 5 in the main text were taken after 2h of continuous operation. This short time interval was taken, as dissolution data showed evidently, that the results in the acidified MEA seemed to be unreliable after that timeframe due to damages in the Pt coating of the stretched metal, used as PTL, as shown in the following figure.

Figure 3 concentrations of Ir, Pt and Ti as a function of time in an MEA experiment with 0.1 M H₂SO₄ circulating in the anode water cycle

The figure shows the concentration of three different metals, Ir, Pt, and Ti, in the anode water cycle in the MEA operated with 0.1 M H₂SO₄ as a function of time. The concentration of Iridium rises in a faster fashion, up to the point where the Ti-concentration in the anode water cycle is shown to rise steadily. We expect that up to this point, Ir concentration is reliable. In contrast, from the point where Ti is exposed to the acid, iridium can deposit on the exposed Ti surfaces, as already suggested by Babic et al. [12].

Finally, the main critics regarding the S-number calculation relies on the way the current density is taken into account. Indeed, the authors should specify that the amount of oxygen evolved as defined in the S-factor is not directly measured (unlike for the amount of dissolved cation), but rather estimated from the anodic current.

The reviewer is right that the amount of evolved oxygen in the S-number calculation is not directly measured but calculated from the measured current density at an estimated 100% faradaic efficiency, which is a good estimation as faradaic efficiency of dissolution is approaching zero.

Thus, we added an explanation to the sentence in l. 131, which is now:

". Figure 1c) shows dissolution stability in both systems, displayed in the S-number metric, a dimensionless descriptor which compares the amount of oxygen evolved, **calculated from the measured current density at an estimated 100% faradaic efficiency towards OER,** with the amount of iridium dissolved ($S - \text{number} = \frac{n(O_2)}{n(Ir)}$) [4]."

Hence, even though the effect of current density is in part retrieved by accounting for it in the S-number calculation, it is not completely suppressed as different current density will mean different local pH as well as difference equilibrium at the interface which can certainly destabilize the catalyst. This could also modify the stability of the catalyst. In order to suppress such uncertainty, measurements at fixed current density (rather than fixed applied potential) can be carried out using different pH, to disentangle both effects.

We agree with the reviewer that measurements at a fixed current density are more promising to exclude local pH effects in OER studies. Thus, we would like to point out that all measurements that contributed to S-numbers' calculation in the whole manuscript were always carried out at fixed current densities and not fixed potentials.

However, we would like to stress that according to Geiger et al. [4], neither current density nor potential have a significant impact on S-numbers of the same materials. Measurements at different current densities or even linear sweep voltammograms of the same material were all in the same order of magnitude regarding the S-number.

Side comments:

What creates such noise in the online ICP-MS measurements when a large amount of dissolved iridium is added?

The ICP-MS presents a highly-precise mass spectrometer, used for elemental trace elements analysis. For measurement purposes, the incoming electrolyte is vaporized by a nebulizer. The vaporized droplets have a stochastic size distribution, and thus, a stochastic distribution of dissolved material.

To ensure reliable data throughout the day of measurements, and minimize any potential drift in the ICP-MS sensitivity, an internal standard is used. The internal standard is an element with a similar mass and ionization energy, which in the case of ICP-MS measurement of Ir is Re. The internal standard is introduced in the device through a second channel alongside the electrolyte from the cell outlet. Counts from the monitored analyte (Ir) are divided by the counts of the internal standard.

When the baseline is set to a higher level by dissolved ions in the electrolyte, the standard deviation of dissolved material per droplet increases relative to the internal standard, which in turn yields a higher noise in the IS-normalized ICP-MS signal.

Unlike stated in the manuscript, Nafion is in fact needed for OER measurements in three electrodes RDE systems testing Ir oxide catalysts in the powder form to ensure the mechanical integrity of the thin catalyst layer upon rotation (while I agree that such Nafion is not needed if the catalyst was prepared by electrodeposition).

In the literature, both RDE measurements with [13] and without [3] Nafion are found for OER catalyst powders. Thus, we believe that our statement regarding Nafion and its necessary role in RDE measurements still stand true.

In addition to the reviewers' comments, we would like to point out that we have corrected a small mistake in the water balance calculation, shown in I56 in the supplementary information.

Also, to comply to the conditions, specified in the editorial policy checklist, statements about calculations of the errorbars, shown in figures 3 and 5, were added to the respective captions.

1. Lædre, S., et al., *Materials for Proton Exchange Membrane water electrolyzer bipolar plates*. International Journal of Hydrogen Energy, 2017. **42**(5): p. 2713-2723.
2. Hsueh, K.-L., et al., *Direct Methanol Fuel Cells*. Electrochemical Technologies for Energy Storage and Conversion, 2012: p. 701-727.
3. Böhm, D., et al., *Efficient OER Catalyst with Low Ir Volume Density Obtained by Homogeneous Deposition of Iridium Oxide Nanoparticles on Macroporous Antimony-Doped Tin Oxide Support*. Advanced Functional Materials, 2019. **30**(1).
4. !!! INVALID CITATION !!! [4].
5. Kasian, O., et al., *The Common Intermediates of Oxygen Evolution and Dissolution Reactions during Water Electrolysis on Iridium*. Angew Chem Int Ed Engl, 2018. **57**(9): p. 2488-2491.
6. !!! INVALID CITATION !!! [5, 6].
7. Pham, C.V., et al., *IrO₂ coated TiO₂ core-shell microparticles advance performance of low loading proton exchange membrane water electrolyzers*. Applied Catalysis B: Environmental, 2020. **269**.
8. !!! INVALID CITATION !!! [8].
9. Zhang, R., et al., *Dissolution/precipitation equilibrium on the surface of iridium-based perovskites as oxygen evolution reaction catalysts in acidic media*. Angew Chem Int Ed Engl, 2019.
10. Bernt, M. and H.A. Gasteiger, *Influence of Ionomer Content in IrO₂/TiO₂ Electrodes on PEM Water Electrolyzer Performance*. Journal of The Electrochemical Society, 2016. **163**(11): p. F3179-F3189.
11. Brightman, E., et al., *In situ characterisation of PEM water electrolyzers using a novel reference electrode*. Electrochemistry Communications, 2015. **52**: p. 1-4.
12. Babic, U., et al., *Understanding the effects of material properties and operating conditions on component aging in polymer electrolyte water electrolyzers*. Journal of Power Sources, 2020. **451**.
13. Fathi Tovini, M., et al., *The Discrepancy in Oxygen Evolution Reaction Catalyst Lifetime Explained: RDE vs MEA Dynamicity within the Catalyst Layer Matters*. Journal of the Electrochemical Society, 2021.

REVIEWERS' COMMENTS

Reviewer #1 (Remarks to the Author):

Thank you for addressing my concerns thoroughly, I can now recommend the publication of the manuscript.

Reviewer #2 (Remarks to the Author):

I would like to thank the authors for their answers to my observations. However, regarding their comments it is well known that HClO₄ is widely used in half cell as liquid electrolyte analogous of a perfluorosulphonate membrane. The reason is essentially regarding the fact that perchlorate anions do not adsorb strongly on the catalysts surface contrary to sulphate anions. Thus, I disagree about the comment that the effect of the selected acid is minimal. If the authors are not happy repeating some tests in HClO₄, I think a thorough discussion on these aspects must be provided in the text.

Similarly, the aspects related to the effect of catalyst loading on stability must be discussed properly because this is another relevant aspect that was not addressed in this work despite this argument is well known to the electrolysis community.

The high flow rate of recirculating water in single cell has an impact that is hard to reproduce in half-cell. This is another aspect that can not be discarded.

Regarding the conclusions, in principle, the proposed sentence is fine. However, this is not a novelty. GDE are widely used in half-cell for testing of practical electrodes and for accurate screening purposes already from the era of phosphoric acid fuel cells. I still do not see any relevant conclusion from this work.

Reviewer #3 (Remarks to the Author):

The authors have very carefully revised the article, being very forthcoming regarding the difficulties encountered when performing such very challenging/sensitive measurements as well as discussing their results in light of what was previously reported/known in the field, which is highly appreciated. Furthermore, I would like to praise once again the quality and seriousness of this study, which is clearly a first in the topic and which will certainly lead to novel findings.

Finally, while I still feel that this article could potentially provide more insights regarding the physical origin for such difference observed between MEA and aqueous configurations, this work is certainly the most in depth analysis in the field. Therefore, I would like to congratulate/thank the authors and warmly recommend this article for publication in Nature Communications.

Rebuttal letter Nature Communications

Ms. Ref. No.: NCOMMS-20-49431A

On the limitations in assessing stability of oxygen evolution catalysts using aqueous model electrochemical cells.

REVIEWER COMMENTS

We thank the editor and reviewers for their time and consideration. Below, we have addressed all comments and questions point by point. The sentences added in the main text are highlighted in yellow.

Reviewer #1 (Remarks to the Author):

Thank you for addressing my concerns thoroughly, I can now recommend the publication of the manuscript.

The authors are happy that they could fully address the reviewers concerns.

Reviewer #2 (Remarks to the Author):

I would like to thank the authors for their answers to my observations. However, regarding their comments it is well known that HClO₄ is widely used in half cell as liquid electrolyte analogous of a perfluorosulphonate membrane. The reason is essentially regarding the fact that perchlorate anions do not adsorb strongly on the catalysts surface contrary to sulphate anions. Thus, I disagree about the comment that the effect of the selected acid is minimal. If the authors are not happy repeating some tests in HClO₄, I think a thorough discussion on these aspects must be provided in the text.

We agree with the reviewer, that there is a difference between the usage of H₂SO₄ and HClO₄ in OER catalyst research. However, the differences in OER activity and stability are not as clear, as the reviewer's question suggests.

First of all, unlike in the oxygen reduction reaction research (ORR) on Pt electrodes, there is no current standard practice in acidic OER electrocatalysis, on which electrolyte should be employed. Both, HClO₄ and H₂SO₄ are used equally in the literature, even today [1]. It is known, as the reviewer states, that sulfate anions adsorb differently to the surface than perchlorate anions [2, 3]. However, the implications on OER electrocatalysis are not as clear, as the reviewer suggests. While some studies show a difference in OER catalyst activity between both electrolytes [3-5], in other works no significant difference is observed [6].

Stability was benchmarked with the dimensionless, current-dependent S-number metric. Based on our observations we can conclude that no grave differences between electrolytes are observed. In this work, IrO_x in aqueous electrolyte displays an S-number of 6×10^4 . In previous work, similar S-numbers for IrO_x under OER conditions are observed in both, sulfuric acid [7, 8] and perchloric acid [9, 10]. Thus, we stand by our statement that stability is not affected significantly by the choice of electrolyte.

To comply with the reviewer's request, we have specified the usage of sulfuric acid electrolyte in the text and its potential role in Ir stability.

To compare the dissolution stability of OER catalysts between MEA and AMS, a commercially available IrO_x catalyst is measured in the aforementioned dedicated MEA system as well as in an SFC-ICP-MS setup operated with 0.1 M H₂SO₄.

Furthermore, to specify that adsorption might happen at the catalyst surface and that the comparison is drawn to measurements in perchloric acid, we have modified the following paragraph.

IrO_x in aqueous electrolyte shows S-numbers between 10⁴ and 10⁵ [9]. This is in line with our measured value of 6×10⁴. Remarkably, the observed S-number of IrO_x in the MEA system exceeds the one observed in aqueous systems by almost five orders of magnitude. Further experiments in AMS were undertaken to unravel the reasons for this behaviour. (Old)

The S-number of IrO_x in the SFC-ICP-MS is 6×10⁴. Although with the employed H₂SO₄ electrolyte a stronger adsorption of anions on the catalysts surface is anticipated [1], the measured S-number in H₂SO₄ is comparable to literature values measured in the non-coordinating HClO₄ electrolyte, which range between 10⁴ and 10⁵ [7-10]. Thus, although the influence of the electrolyte anion cannot be fully ruled out, its role in the stability of IrO_x is minor. Remarkably, the observed S-number of IrO_x in the MEA system exceeds the one observed in aqueous systems by almost five orders of magnitude. Further experiments in AMS were undertaken to unravel the reasons for this behaviour. (New)

Similarly, the aspects related to the effect of catalyst loading on stability must be discussed properly because this is another relevant aspect that was not addressed in this work despite this argument is well known to the electrolysis community.

The high flow rate of recirculating water in single cell has an impact that is hard to reproduce in half-cell. This is another aspect that can not be discarded.

As demonstrated, catalyst dissolution is, at least in the aqueous system, independent of loading. The results seem to be scalable at least in the aqueous system. However, we agree with the reviewer that, due to very high loadings, and the different complexity of both systems, loading and flow rate might play a role in MEA. Although we already concluded this, to make it more clear, we extended a sentence in the section "System breakdown of the dissolution discrepancy between AMS and MEA".

These differences cannot be related to a sole factor. Loading, flow rate, dissolved species, timescale or Nafion content, although not relevant factors in AMS, might play a role in MEA and have to be addressed in a follow-up study.

Regarding the conclusions, in principle, the proposed sentence is fine. However, this is not a novelty. GDE are widely used in half-cell for testing of practical electrodes and for accurate screening purposes already from the era of phosphoric acid fuel cells. I still do not see any relevant conclusion from this work.

While GDE testing might not be a novelty for most electrochemical applications, for the screening of OER catalysts it is. The first test of OER catalysts in a GDE setup was published only weeks ago [11], and it is still not clear if this test is representative, as we believe that significant improvements of the modern

GDEs used in the fuel cell research are necessary. To provide relevant data there is still a way to go. Thus, we believe, that our conclusions are valid.

Reviewer #3 (Remarks to the Author):

The authors have very carefully revised the article, being very forthcoming regarding the difficulties encountered when performing such very challenging/sensitive measurements as well as discussing their results in light of what was previously reported/known in the field, which is highly appreciated. Furthermore, I would like to praise once again the quality and seriousness of this study, which is clearly a first in the topic and which will certainly lead to novel findings. Finally, while I still feel that this article could potentially provide more insights regarding the physical origin for such difference observed between MEA and aqueous configurations, this work is certainly the most in depth analysis in the field. Therefore, I would like to congratulate/thank the authors and warmly recommend this article for publication in Nature Communications.

The authors thank the reviewer for the appreciation expressed to the work carried out in this manuscript. We hope as well that this research will open the door to novel findings.

1. Arminio-Ravelo, J.A., et al., *Electrolyte effects on the electrocatalytic performance of iridium-based nanoparticles for oxygen evolution in rotating disc electrodes*. Chemphyschem, 2019.
2. Fonseca, I.T.E., M.I. Lopes, and M.T.C. Portela, *A comparative voltammetric study of the ir/h₂so₄ and ir/hclo₄ aqueous interfaces*. Journal of Electroanalytical Chemistry, 1996. **415**(1-2): p. 89-96.
3. Ganassin, A., et al., *Non-covalent interactions in water electrolysis: influence on the activity of Pt(111) and iridium oxide catalysts in acidic media*. Phys Chem Chem Phys, 2015. **17**(13): p. 8349-55.
4. Strickler, A.L., D. Higgins, and T.F. Jaramillo, *Crystalline Strontium Iridate Particle Catalysts for Enhanced Oxygen Evolution in Acid*. ACS Applied Energy Materials, 2019. **2**(8): p. 5490-5498.
5. Diaz-Morales, O., et al., *Electrochemical and spectroelectrochemical characterization of an iridium-based molecular catalyst for water splitting: turnover frequencies, stability, and electrolyte effects*. J Am Chem Soc, 2014. **136**(29): p. 10432-9.
6. Alia, S.M. and G.C. Anderson, *Iridium Oxygen Evolution Activity and Durability Baselines in Rotating Disk Electrode Half-Cells*. Journal of The Electrochemical Society, 2019. **166**(4): p. F282-F294.
7. da Silva, G.C., et al., *Dissolution Stability: The Major Challenge in the Regenerative Fuel Cells Bifunctional Catalysis*. Journal of The Electrochemical Society, 2018. **165**(16): p. F1376-F1384.
8. Silva, G.C., et al., *Oxygen Evolution Reaction on Tin Oxides Supported Iridium Catalysts: Do We Need Dopants?* ChemElectroChem, 2020. **7**(10): p. 2330-2339.
9. Geiger, S., et al., *The stability number as a metric for electrocatalyst stability benchmarking*. Nature Catalysis, 2018. **1**(7): p. 508-515.
10. Pham, C.V., et al., *IrO₂ coated TiO₂ core-shell microparticles advance performance of low loading proton exchange membrane water electrolyzers*. Applied Catalysis B: Environmental, 2020. **269**.
11. Schröder, J., et al., *The Gas Diffusion Electrode Setup as Straightforward Testing Device for Proton Exchange Membrane Water Electrolyzer Catalysts*. JACS Au, 2021.